# Determining zebrafish dorsal organizer size by a negative feedback loop between canonical/non-canonical Wnts and Tlr4/NFκB

Juqi Zou [1], Satoshi Anai [2], Satoshi Ota [3], Shizuka Ishitani [1], Masayuki Oginuma [1] & Tohru Ishitani [1,4] ✉

In vertebrate embryos, the canonical Wnt ligand primes the formation of dorsal organizers that govern dorsal-ventral patterns by secreting BMP antagonists. In contrast, in *Drosophila* embryos, Toll-like receptor (Tlr)-mediated NFκB activation initiates dorsal-ventral patterning, wherein Wnt-mediated negative feedback regulation of Tlr/NFκB generates a BMP antagonist-secreting signalling centre to control the dorsal-ventral pattern. Although both Wnt and BMP antagonist are conserved among species, the involvement of Tlr/NFκB and feedback regulation in vertebrate organizer formation remains unclear. By imaging and genetic modification, we reveal that a negative feedback loop between canonical and non-canonical Wnts and Tlr4/NFκB determines the size of zebrafish organizer, and that Tlr/NFκB and Wnts switch initial cue and feedback mediator roles between *Drosophila* and zebrafish. Here, we show that canonical Wnt signalling stimulates the expression of the non-canonical Wnt5b ligand, activating the Tlr4 receptor to stimulate NFκB-mediated transcription of the Wnt antagonist frzb, restricting Wnt-dependent dorsal organizer formation.

Animals display various body plans comprising various anatomical axes such as the dorsal-ventral (DV) and anterior-posterior (AP) axes. The establishment of body axes is one of the most fundamental events in the development of multicellular organisms. Since its discovery, the Spemann-Mangold organizer[1], a group of cells that initiate DV axis formation in the amphibian embryo, has been intensively studied in a variety of model animals. It was found that organizer formation is induced by Wnt/β-catenin signaling. Canonical Wnt ligands, such as zebrafish Wnt8a and *Xenopus* Wnt11, activate β-catenin signaling, specifically in the dorsal embryonic region, thereby stimulating the formation of the dorsal organizer[2–6]. The dorsal organizer secretes the

BMP antagonist Chordin into the ventral region; then, Chordin inhibits BMP-dependent ventral specification[7–9]. Thus, Wnt/β-catenin signaling initiates DV axis formation through Chordin/BMP in vertebrates. On the other hand, in *Drosophila*, the DV axis formation is initiated by Toll-like receptor (Tlr)/NFκB signaling[10–13]. Spätzle (Spz) ligands are proteolytically cleaved, specifically in the ventral-most region, which then activates the Tlr homolog (Toll). Activated Tlr stimulates the degradation of the IκB homolog, allowing nuclear translocation of the NFκB family of transcription factors and consequent transcriptional activation of genes for ventral specification[14,15]. Concurrently, NFκB also induces the expression of the Wnt family of extracellular protein

[1]Department of Homeostatic Regulation, Division of Cellular and Molecular Biology, Research Institute for Microbial Diseases, Osaka University, Suita, Osaka 565-0871, Japan. [2]Yuuai Medical Center, Tomigusuku, Okinawa 901-0224, Japan. [3]Genome Science Division, Research Center for Advanced Science and Technology, The University of Tokyo, Komaba 4-6-1, Meguro-ku, Tokyo 153-8904, Japan. [4]Center for Infectious Disease Education and Research (CiDER), Osaka University, Suita, Osaka 565-0871, Japan. ✉e-mail: ishitani@biken.osaka-u.ac.jp

WntD, functioning as an antagonist to attenuate Tlr/NFκB signaling[16,17]. This Wnt-mediated negative feedback regulation of Tlr/NFκB signaling is responsible for the precise size of the ventral embryonic region. Moreover, Tlr/NFκB signaling represses the BMP homolog (Dpp)-mediated dorsal specification by inducing the expression of the Chordin homolog (Sog)[18,19]. Thus, Tlr/NFκB signaling initiates *Drosophila* DV axis formation through the regulation of Wnt, Chordin, and BMP homologs. Taken together, Wnt, Chordin, and BMP are conserved mediators of DV axis formation in vertebrates and *Drosophila*.

Since the Wnt/Chordin/BMP system is evolutionarily conserved and *Drosophila* Tlr/NFκB signaling initiates DV axis formation, it is expected that vertebrate Tlr/NFκB signaling might also be involved in DV axis formation. Overexpression of NFκB family genes reportedly inhibits dorsal formation in *Xenopus laevis*[20,21]. Other studies have shown that injection of *Drosophila* Spätzle and Tlr homolog into *Xenopus* embryos induced a secondary axis[22], and that overexpression of IκB (inhibition of NFκB) blocked *Xenopus* dorsal formation[23]. While these findings indicate that Tlr/NFκB signaling may respectively function as a negative or positive regulator of dorsal formation, these overexpression studies remain controversial. Furthermore, although large-scale screening for isolating zebrafish mutants with dorsoventral patterning defects has been performed[24], Tlr/NFκB signaling-related factors have not been isolated. Thus, the function and regulation of endogenous Tlr/NFκB signaling during vertebrate DV axis formation remain unclear.

The negative feedback loop plays an important role in axis formation and size control. For example, a Sizzled-mediated BMP-Chordin feedback loop is required for correct DV patterning and embryonic size control[25]. Feedback regulation between Wnt and its secreted inhibitor Dkk1 contributes to size control of sensory organs[26]. The Wnt antagonist Sfrp1-mediated negative feedback regulation of Wnt/β-catenin signaling is essential for the development of a normal-sized heart muscle[27]. Because the dorsal organizer is the signaling center priming axis formation, organizer size should be properly controlled, raising the possibility that Wnt/β-catenin signaling, the organizer-inducer, may be restricted by negative feedback regulation. However, this mechanism is poorly understood.

In this study, we examined the function and regulation of endogenous Tlr and NFκB during zebrafish DV axis formation, using a combination of in vivo reporter analysis, CRISPR/Cas9-mediated knockout, and morpholino knockdown. We show that during the initiation of dorsal organizer formation, Wnt/β-catenin signaling stimulates the activation of the NFκB homolog Rel through Toll-like receptor 4 (Tlr4), specifically in the dorsal embryonic tissue. Surprisingly, the non-canonical Wnt5 ligand mediates β-catenin-dependent Tlr4/Rel activation. Activated Rel then stimulates the transcription of a Wnt antagonist, frizzled-related protein (*frzb*), thereby restricting the Wnt/β-catenin-active area and dorsal organizer size. Thus, Wnt5-Tlr4/NFκB-mediated indirect negative feedback regulation of Wnt/β-catenin signaling determines the precise size of zebrafish dorsal organizer.

## Results

### NFκB activation in the dorsal region of zebrafish embryos
To clarify the spatiotemporal pattern of NFκB activity, we generated a new NFκB reporter, NFκB-tkP:dGFP (Fig. 1a). We confirmed that activation of NFκB stimulated NFκB-tkP:dGFP activity in human HEK293 cells (Supplementary Fig. 1a) and then generated stable transgenic zebrafish lines carrying a single copy of NFκB-tkP:dGFP (Supplementary Fig. 1b). NFκB-tkP:dGFP activity in transgenic fish was detected at 3.7 hours-post-fertilization (hpf) (Fig. 1b), indicating that the reporter gene was zygotically activated. The reporter expression gradually accumulated to the dorsal margin of the blastoderm, which corresponds to the future dorsal organizer, from the dome stage (4.3 hpf), and completely localized in the dorsal region at the 50% epiboly stage

(5.3 hpf) (Fig. 1b, c). These results suggest that NFκB functions in dorsal organization.

### Rel/NFκB negatively regulates dorsal organizer formation
To test whether NFκB is involved in dorsal organizer formation, we overexpressed the zebrafish IκB homolog *iκbab* to block NFκB activity in early zebrafish embryos (Fig. 1d). Overexpression of *iκbab* induced expansion of the organizer area and dorsal tissue, marked by the expression of *dharma* and *chordin*, respectively[7,9,28–30], in early embryos (Fig. 1e), resulting in class 2–3 (C2–3) dorsalizations[31,32], with a significant loss of ventral tail fin in larvae (Fig. 1f). These results suggest that NFκB negatively regulates dorsal specification.

Next, we investigated which NFκB regulates dorsal cell fate. In zebrafish, the NFκB family comprises five members: Rel (mammalian c-Rel homolog), Rela, Relb, NFκB1, and NFκB2 (Supplementary Fig. 2a). We focused on Rel because it is the most homologous to *Drosophila* Dorsal, with high levels of *rel* transcripts being detected in early embryos (Supplementary Fig. 2a, b). *rel* overexpression dramatically activated the NFκB reporter, narrowed the size of organizer and dorsal tissue, and reduced expression levels of the organizer marker *dharma* and the dorsal tissue marker *chordin* in early embryos (Fig. 1g, h and Supplementary Fig. 2c). This induced the ventralized V1–4 phenotype in most larvae, characterized by the loss of dorsoanterior structures and expanded ventral tissues[32] (Fig. 1i), which indicates that Rel may possibly inhibit the formation of dorsal organizer and dorsal tissue. To confirm this hypothesis, we used antisense morpholino (MO) to knockdown *rel*, blocking the translation of *rel* mRNA (Supplementary Fig. 2d). Injection of *rel* MO dramatically reduced NFκB-tkP:dGFP activity (Fig. 2a) and induced the expansion of dorsal tissues, marked by the expression of *dharma* and *chordin*, respectively (Fig. 2b and Supplementary Fig. 2e), and the reduction of ventral tissues, marked by *vent*[33] (Fig. 2c), resulting in class 2 (C2) dorsalization (Fig. 2d). In addition, morpholino-resistant *rel* mRNA rescued *rel* MO-induced dorsal expansion (Supplementary Fig. 2f). These results suggest that Rel restricts dorsal organizer formation and consequent dorsal specification. Although another NFκB gene, *rela*, is also expressed in early embryos (Supplementary Fig. 2b), *rela* MO[34] did not affect the expression of dorsal marker genes (Fig. 2b and Supplementary Fig. 2g) nor enhance *rel* MO-induced increase of dorsal gene expression (Supplementary Fig. 2g). This suggests that Rel, but not Rela, is the main NFκB that acts in early zebrafish dorsal organizer formation.

### Upregulation of *rela* compensates the genetic loss of *rel*
To confirm the role of *rel*, we generated a *rel* mutant by CRISPR/Cas9-mediated knockout, which has a frameshift in the Rel homology domain, leading to early termination of translation (Supplementary Fig. 2h). We also confirmed that *rel* expression was dramatically decreased in the maternal-zygotic *rel* mutants (MZ*rel*) (Supplementary Fig. 2i). Unexpectedly, MZ*rel* embryos exhibited no gross morphological defects (Fig. 2d, top-right panel). Consistent with this, there were no significant differences in the expression of dorsal marker genes between wild-type (WT) and MZ*rel* embryos (Fig. 2b and Supplementary Fig. 2e). A previous study showed that the lack of a mutant phenotype is due to the upregulation of either gene paralogs or genes with sequence homology[35–37]. We found that *rela* was significantly upregulated in MZ*rel*, but not *rel* morphants (Fig. 2e, f). Moreover, injection of *rela* MO enhanced the expression of dorsal markers (Fig. 2b) and induced dorsalized phenotypes in MZ*rel* embryos, but not in WT embryos (Fig. 2d, f). These results indicate that the upregulation of *rela* compensates for the genetic loss of *rel* in mutants but not *rel* morphants (Fig. 2f). Notably, MZ*rel* is *rel* MO-resistant, with no obvious defects after *rel* MO injection (Fig. 2b, d, f and Supplementary Fig. 2e), indicating that MZ*rel* is a null mutant and that *rel* MO specifically inhibits the function of *rel*. Therefore, we used this specific morpholino to investigate the functions of Rel.

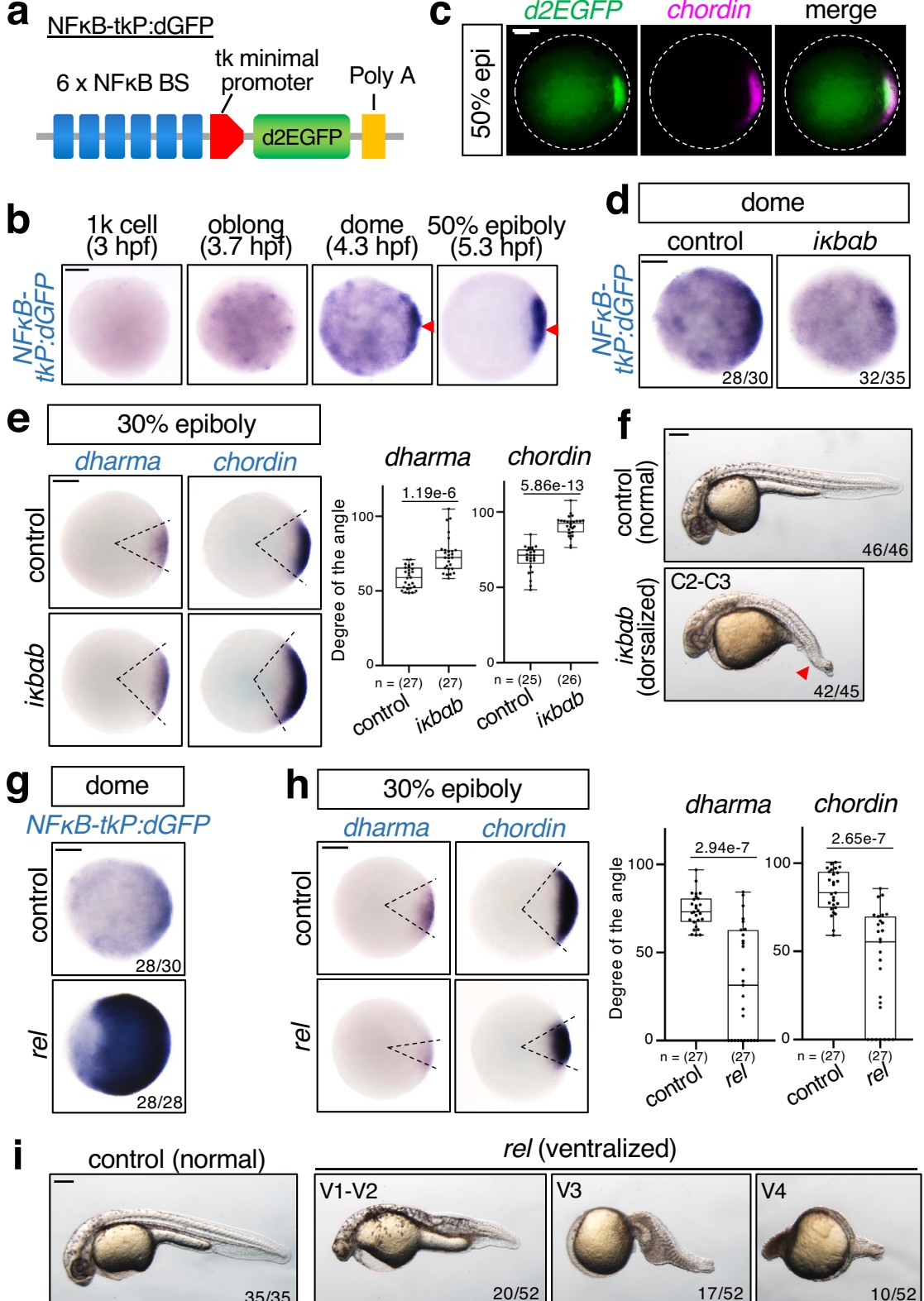

**Rel inhibits Wnt signaling through activating Frzb**

Given that Wnt/β-catenin initiates the formation of the dorsal organizer, we investigated whether Rel regulates Wnt/β-catenin signaling. Injection of *rel* MO resulted in a significant increase in *axin2* and *sp5l*, the specific targets of Wnt/β-catenin activity[38] (Fig. 3a, b and Supplementary Fig. 3a). Consistently, *rel* overexpression significantly decreased the expression of Wnt target genes (Fig. 3c, d). However, *rel*

MO did not affect the expression of Nodal (*ndr1* and *ndr2*)[39] and BMP signaling target genes (*foxi1* and *bambia*)[40,41], which are also involved in DV axis formation[42], at the early stage of dorsal organizer induction (Supplementary Fig. 3b). These results suggest that Rel selectively inhibits Wnt/β-catenin signaling during organizer formation.

Next, we investigated how Rel negatively regulated Wnt/β-catenin signaling. As Rel functions as a transcription factor with a

**Fig. 1 | Rel/NFκB negatively regulates dorsal organizer formation. a** Schematic diagrams of NFκB-tkP:dGFP reporter. NFκB BS: consensus sequence of the NFκB-binding element. PolyA: SV40 polyadenylation sequence. **b, c** NFκB reporter is activated in the dorsal region of early zebrafish embryos. **b** Whole-mount in situ hybridization (WISH) for *dGFP* at the indicated stage in NFκB-tkP:dGFP-transgenic embryos. The *dGFP*-expressing dorsal regions are indicated with red arrowheads. Scale bar = 200 µm. **c** Double fluorescent in situ hybridization (FISH) for *dGFP* (green) and *chordin* (dorsal marker, magenta) in NFκB-tkP:dGFP-transgenic embryos. Animal views with dorsal to the right. Scale bar = 200 µm. **d–f** Inhibition of NFκB activity by *iκbab* leads to expansion of organizer size and dorsal tissue. **d** WISH for *dGFP* in NFκB-tkP:dGFP-transgenic embryos, (**e**) *dharma* (organizer marker) and *chordin* in embryos injected with control (mKO2) or *iκbab* mRNA at indicated stages. Animal views. Scale bar = 200 µm. Box plots of the angle of marker genes show first and third quartile, median is represented by a line, whiskers indicate the minimum and maximum. Each dot represents one embryo. *P*-values

from unpaired two-tailed *t*-tests are indicated. **f** Phenotypes of larvae injected with control (mKO2) or *iκbab* mRNA at 27 hpf. The strength of dorsalization was scored using the C1–5 classification scheme. Lateral views with anterior to the left. The loss of ventral tail fin (a typical dorsalization phenotype) is indicated with red arrowheads. Scale bar = 200 µm. **g–i** *rel* over-expression leads to reduction of organizer size and dorsal tissue. **g** WISH for *dGFP* in NFκB-tkP:dGFP-transgenic embryos, (**h**) *dharma* and *chordin* in embryos injected with control (mKO2) or *rel* mRNA at indicated stages. Scale bar = 200 µm. Box plots of the angle of marker genes show first and third quartile, median is represented by a line, whiskers indicate the minimum and maximum. Each dot represents one embryo. *P*-values from unpaired two-tailed *t*-tests are indicated. **i** Phenotypes of 27 hpf larvae injected with control (mKO2) or *rel* mRNA. The strength of ventralization was scored. Lateral views with anterior to the left. Scale bar = 200 µm. Source data are provided as a Source Data file.

transactivation domain, we hypothesized that Rel stimulates the expression of Wnt/β-catenin signaling inhibitors. Therefore, we examined the expression of Wnt antagonists in early zebrafish embryos[6,43]. Knockdown of *rel* significantly reduced the expression of the Wnt antagonist *frzb* in the dorsal margin but had no significant effect on the expression of other Wnt antagonists (Fig. 4a, b and Supplementary Fig. 3a, b), whereas *rel* overexpression enhanced *frzb* expression (Supplementary Fig. 3c). These outcomes suggest that Rel selectively activates Frzb expression. Consistent with a previous study[6], we confirmed that *frzb* knockdown using MO induced the upregulation of the Wnt target gene *axin2* and dorsal expansion phenotypes, whereas overexpression of *frzb* induced the opposite effects (Supplementary Fig. 4a–c). Moreover, *rel* MO-induced Wnt/β-catenin activation and dorsal expansion were restored by co-injection with *frzb* mRNA (Fig. 4c, d), suggesting that Rel inhibited Wnt/β-catenin signaling by inducing Frzb expression. It is worth noting that overexpression of *sfrp1a* also rescued *rel* MO-induced activation of Wnt signaling and dorsal expansion (Supplementary Fig. 4d, e). This suggests that antagonizing Wnt ligand activity is important for size control of the dorsal organizer.

We also discovered three elements that possess binding potential to Rel 2 kb upstream of the zebrafish *frzb* gene (Fig. 4e), using the NFκB-binding elements search tool[44]. Among these, the element that was nearest to *frzb* gene (−246 to −236 bp) had the strongest binding potential to Rel (Fig. 4e). To examine whether Rel directly activates *frzb* transcription, we generated a *frzb*:luc reporter in which a 500 bp promoter sequence, including the potential Rel-binding element, was fused with the luciferase gene and the Rel-binding element-deleted mutant (MT) reporter *frzb*:luc MT (Fig. 4e). Embryos injected with *frzb*:luc, but not with *frzb*:luc MT, expressed luciferase mRNA in the dorsal organizer region, which was confirmed by co-injection with the organizer reporter pDharma-GFP[45] (Fig. 4f). Knockdown of *rel* decreased the activity of *frzb*:luc reporter, whereas overexpression of *rel* activated it (Fig. 4f and Supplementary Fig. 3d). These results suggest that Rel directly activated *frzb* promoter through the Rel-binding element.

### Tlr4 activates NFκB to stimulate *frzb* expression

Next, we investigated NFκB activation. Because Tlr/NFκB signaling determines *Drosophila* DV axis, we investigated whether Tlr mediates NFκB activation in early zebrafish embryos. We focused on Tlr4 because it is highly expressed among Tlrs in early zebrafish embryos[46]. We confirmed that *tlr4* was expressed in early zebrafish embryos (Supplementary Fig. 5a). Similar to *rel* knockdown, injection of a dominant-negative mutant of Tlr4 (Tlr4 DN)[47] or treatment with the Tlr4-specific inhibitor TAK-242[48,49] induced a reduction in NFκB reporter activity and *frzb* expression and the upregulation of the Wnt target gene *axin2*, the dorsal organizer gene *dharma*, and the dorsal marker *chordin* (Fig. 5a, b and Supplementary Fig. 5b–d) and

consequent dorsalized phenotype (Fig. 5c and Supplementary Fig. 5e). Furthermore, forced activation of Tlr4 by injection of lipopolysaccharide (LPS), which is an exogenous Tlr4 ligand derived from the cell wall of gram-negative bacteria[50], induced a ventralized phenotype at 27 hpf (Fig. 5d). LPS significantly activated the NFκB reporter and enhanced *frzb* expression, which was restored by either Tlr4 DN injection or TAK-242 treatment (Fig. 5e and Supplementary Fig. 5f). Moreover, LPS injection decreased the expression of the Wnt target gene *axin2* and the dorsal marker genes *dharma* and *chordin* (Fig. 5f and Supplementary Fig. 5g). These results suggest that Tlr4 activates NFκB to stimulate *frzb*-mediated restriction of dorsal organizer formation.

### β-catenin stimulates Wnt5-mediated Tlr4/NFκB activation

Since Wnt/β-catenin signaling and NFκB are activated in the dorsal region, it is possible that Wnt/β-catenin also regulates NFκB signaling. Interestingly, forced activation of Wnt/β-catenin signaling by a constitutively active β-catenin mutant (β-cat CA) dramatically enhanced NFκB reporter activity and *frzb* expression, which was reversed by co-injection with Tlr4 DN (Fig. 6a). In contrast, knockdown of the β-catenin homolog *ctnnb2* by translation-blocking MO[51] decreased both NFκB reporter activity and *frzb* expression (Fig. 6b). These results suggest that Wnt/β-catenin signaling activates NFκB through Tlr4. Consistent with this, the Wnt/β-catenin reporter OTM:d2EGFP[52] was initially activated at 3.7 hpf, which is earlier than the NFκB reporter and NFκB target gene *frzb* (Fig. 6c). Thus, Wnt/β-catenin and Tlr4/NFκB form an indirect negative feedback loop.

As *tlr4* genes are ubiquitously expressed, we hypothesized that Wnt/β-catenin may stimulate the expression of a Tlr4 ligand in the dorsal region. A recent study showed that the non-canonical Wnt ligand Wnt5 is a Tlr4 ligand in human myeloid cell cultures[53]. Moreover, similar to Tlr4/NFκB, Wnt5 negatively regulates dorsal formation by blocking Wnt/β-catenin signaling[54–57]. In particular, the endogenous *wnt5b* gene negatively regulates zebrafish dorsal organizer formation. Therefore, we tested whether β-catenin activates Tlr4/NFκB through Wnt5b. We found that *wnt5b* was specifically expressed in the dorsal region (Supplementary Fig. 6a) and significantly activated after β-cat CA injection, whereas the expression of another Wnt5 gene, *wnt5a*, was not detected (Fig. 6d), indicating that *wnt5b* is a β-catenin target gene in early zebrafish embryos. Furthermore, knockdown of Wnt5b using *wnt5b* translation-blocking MO[58] decreased the NFκB activity significantly (Fig. 6e) and reversed β-cat CA-induced NFκB activation (Fig. 6f), suggesting that β-catenin activates Tlr/NFκB signaling through Wnt5b. Consistent with this idea, overexpression of *wnt5b* mRNA significantly enhanced NFκB reporter and *frzb* expression which were restored by either Tlr4 DN or *rel* MO injection (Fig. 6g and Supplementary Fig. 6b). In addition, Wnt5b overexpression also inhibited dorsal organizer formation (Fig. 6h). Taken

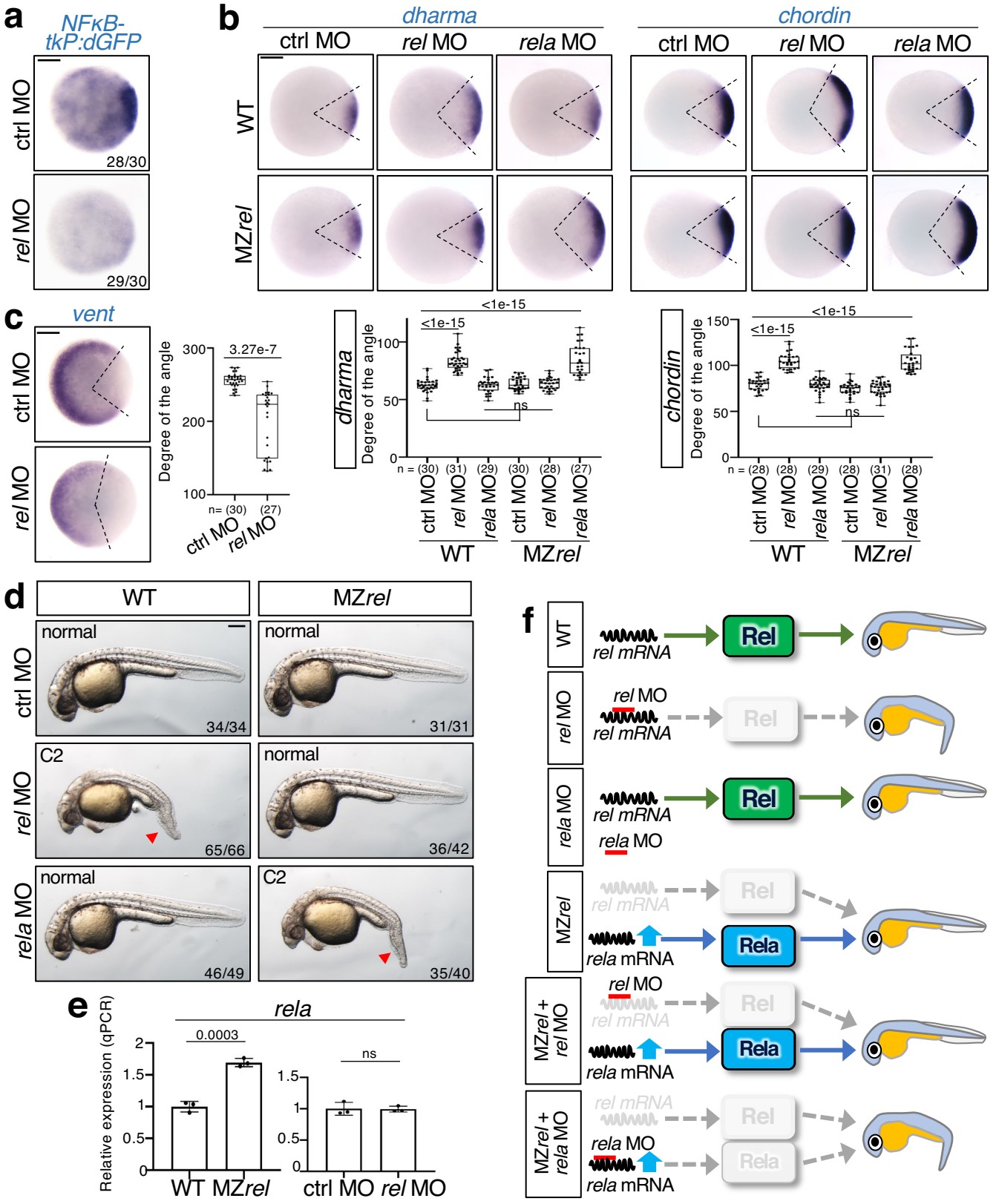

together, Wnt/β-catenin signaling appears to activate Tlr4/NFκB through the non-canonical Wnt ligand, Wnt5b.

## Discussion

In this study, we demonstrate that the Tlr4/NFκB-mediated negative feedback regulation of Wnt/β-catenin signaling determines the precise size of the zebrafish dorsal organizer. In early zebrafish embryos, Wnt/β-catenin signaling stimulates the transcription of the non-canonical Wnt5b ligand, which activates the NFκB protein Rel through Tlr4 in the dorsal region. Rel stimulates the transcription of the Wnt antagonist *frzb*, thereby restricting the Wnt/β-catenin-active area and dorsal organizer size. Similar to *Drosophila*, zebrafish determine their

**Fig. 2 | Specific *rel* MO reveals that Rel restricts dorsal organizer formation.** **a** Rel is a main NFκB acting in early zebrafish embryos. WISH for *dGFP* in NFκB-tkP:dGFP-transgenic embryos injected with control MO (ctrl MO) or *rel* MO at dome stage, animal view. Scale bar = 200 μm. **b–d** *rel* MO leads to expansion of organizer size and dorsal tissue in wild-type (WT) embryos whereas *rela* compensates the loss of *rel* in MZ*rel* embryos. WISH for (**b**) *dharma*, *chordin*, and (**c**) *vent* (ventral marker) in WT and maternal-zygotic *rel* mutants (MZ*rel*) embryos injected with ctrl MO, *rel* MO or *rela* MO at 30% epiboly. Animal views with dorsal to the right. Scale bar = 200 μm. Box plots of the angle of marker genes show first and third quartile, median is represented by a line, whiskers indicate the minimum and maximum. Each dot represents one embryo. *P*-values from two-tailed one-way ANOVAs with Sidak correction are indicated. ns: not significant (*p* > 0.05). **d** Representative pictures of 27 hpf WT and MZ*rel* larvae injected with ctrl MO, *rel* MO or *rela* MO. The strength of dorsalization was scored. The loss of ventral tail fin is indicated with red arrowheads. Lateral views with anterior to the left. Scale bar = 200 μm. **e** *rela* is upregulated in MZ*rel* but not *rel* morphants. qPCR analysis for expression of *rela* in MZ*rel* or *rel* morphants at sphere stage. Normalized values are shown as means ± SEM. *n* = 3, biologically independent samples. *P*-values from unpaired two-tailed t-tests are indicated. **f** Model of the mechanism of *rela*-mediated genetic compensation and the specific inhibition by *rel* MO. Source data are provided as a Source Data file.

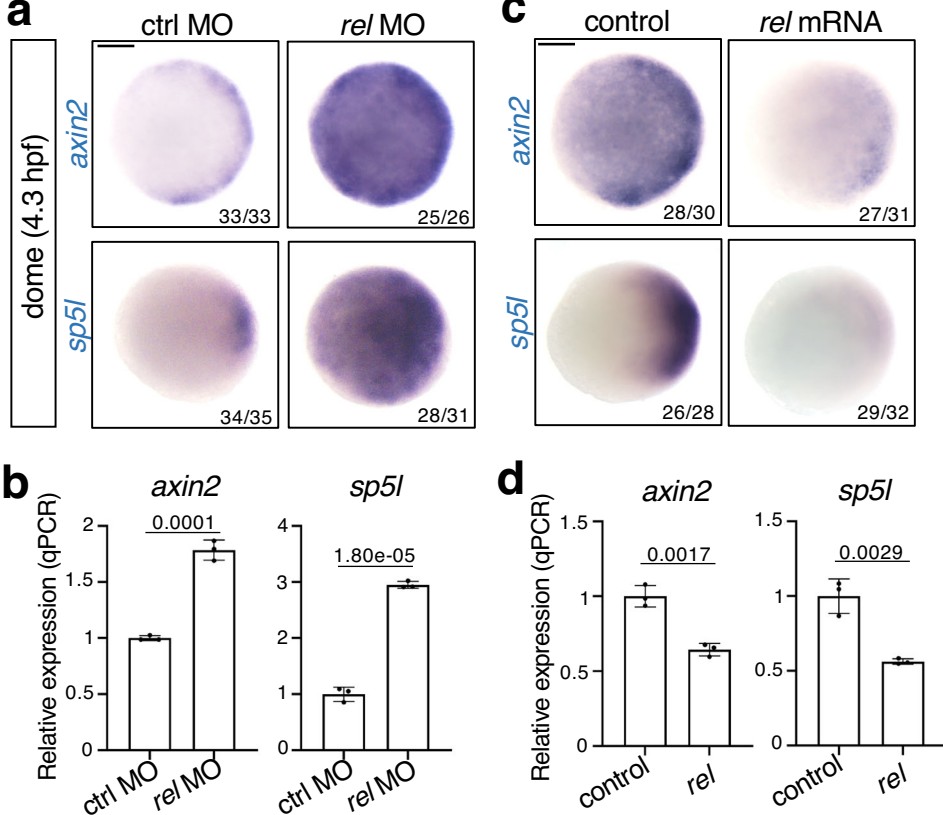

**Fig. 3 | Rel negatively regulates Wnt/β-catenin signaling.** Wnt/β-catenin signaling is enhanced by *rel* knockdown (**a**, **b**) and decreased by *rel* overexpression (**c**, **d**) at the dome stage. WISH (**a**, **c**) and qPCR analysis (**b**, **d**) of *axin2* and *sp5l* in dome-stage WT embryos injected with ctrl MO, *rel* MO, control (mKO2) or *rel* mRNA. In **a** and **c**, animal views are shown. Scale bar = 200 μm. In **b** and **d**, normalized values are shown as mean ± SEM. *n* = 3 biologically independent samples. *P*-values from unpaired two-tailed *t*-tests are indicated. Source data are provided as a Source Data file.

embryonic DV axis through negative feedback regulation between Tlr/NFκB and Wnt. Interestingly, the roles of these factors appear to be switched between these two species. Tlr/NFκB acts as the initial cue of DV axis formation in *Drosophila* and as a feedback mediator in zebrafish, whereas Wnt functions as the initial cue in zebrafish and as a feedback mediator in *Drosophila* (Fig. 7).

To precisely generate embryonic tissue of specific size, morphogen signaling is adjusted for correct distribution. Negative feedback regulation measures tissue size and buffer variability. For example, the sizes of sensory organs and heart muscles are correctly determined by negative feedback regulation between Wnt and its secreted antagonists Dkk1 and Sfrp[26,27]. The dorsal organizer is formed by Wnt and expresses secreted Wnt antagonists, including Frzb[6,59], which raises the possibility that negative feedback regulation between Wnt and antagonists may be involved in organizer size control. However, such feedback regulation remains unreported, and the mechanisms by which Frzb expression is controlled in organizer remain unclear.

In this study, we show that a previously unidentified "indirect" negative feedback loop, consisting of Wnt, Tlr4/NFκB, and Frzb, restricts the size of the dorsal organizer. Interestingly, Wnt indirectly activates Frzb through Tlr4/NFκB, whereas the well-studied Wnt negative feedback regulator Dkk1 is a direct Wnt target gene[60,61]. The significance of "indirect" feedback lies in the fact that while it is known that a "direct" autoregulatory feedback loop can shorten the response time of a network, indirect feedback via intermediate genes can generate time delay[62–64]. If Wnt signaling is inhibited through direct negative feedback during organizer induction, Wnt signaling would be immediately shut off and would not be able to form an organizer of an adequate size. Indirect negative feedback delays the timing of Wnt inhibition, which may ensure an adequate duration for Wnt diffusion and consequent formation of a properly sized organizer (Supplementary Fig. 7).

Since the discovery of Tlr/NFκB (Toll/Dorsal) signaling as an initiator of *Drosophila* DV axis formation[10–13], the involvement of Tlr/

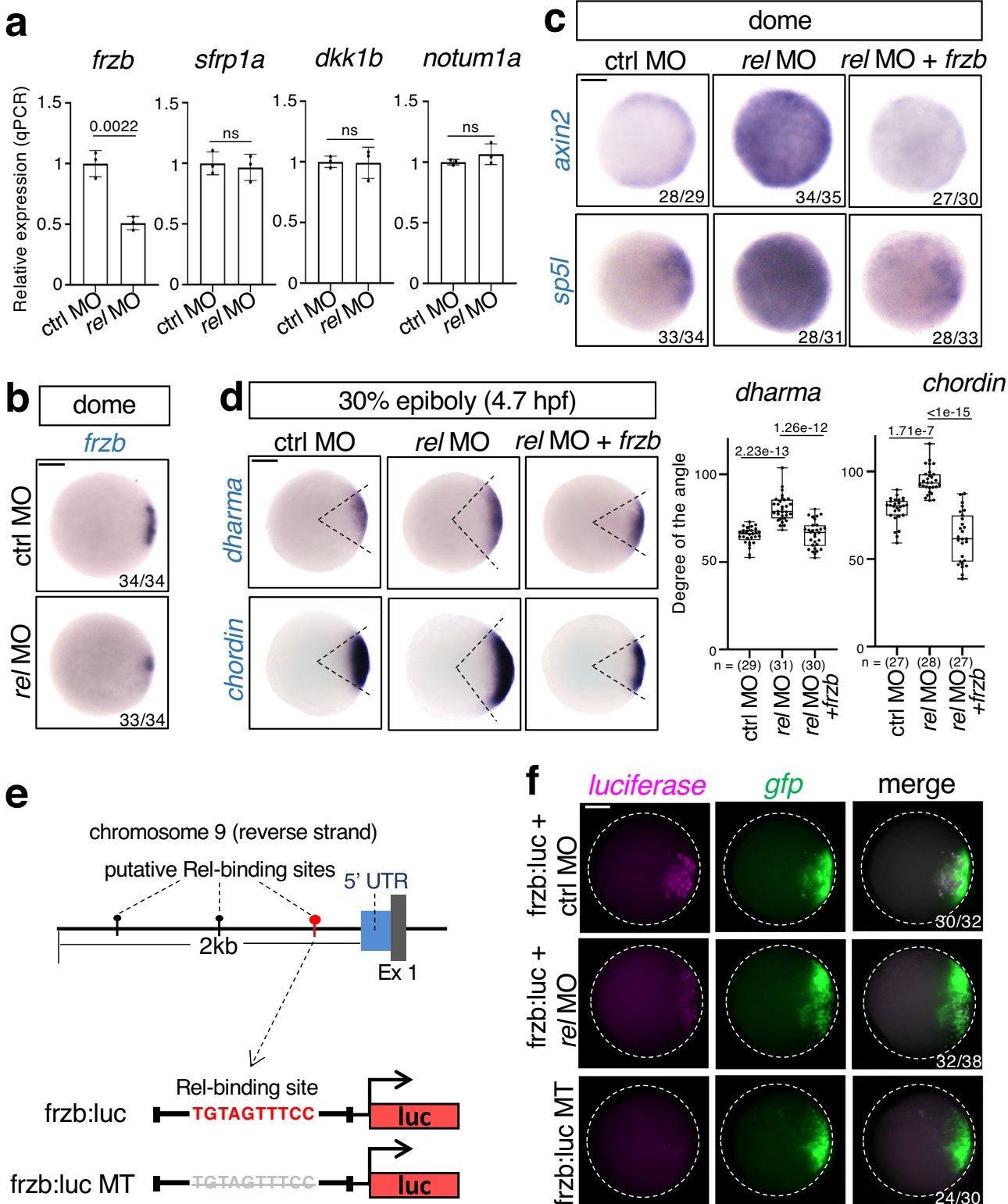

NFκB signaling in vertebrate DV axis formation has been expected. However, no related factors were found in large-scale screening for isolating zebrafish mutants with dorsoventral patterning defects[24]. Furthermore, previous *Xenopus* studies showed that dorsal formation can be blocked by overexpression not only of NFκB[20,21] but also of IκB[23], indicating that these overexpression analyses are controversial and potentially insufficient to prove the contribution of these factors to axis formation. Moreover, while mutant mice with disrupted key components of the Tlr/NFκB pathway demonstrated immune deficiency, no significant defects in body axis formation were observed[65–68]. Genetic compensation can reportedly buffer the organism against gene loss that would otherwise be deleterious to survival, whereas morpholino-mediated knockdown blocked the target gene immediately and induced no genetic compensation[35–37], suggesting that both knockout and knockdown analyses are required to clarify endogenous gene function.

Here, we show that genetic compensation masks the roles of Rel, a member of the NFκB family, in early zebrafish embryos, and we clarify

**Fig. 4 | Rel directly activates a secreted Wnt antagonist Frzb to restrict Wnt/β-catenin-mediated dorsal organizer formation. a, b** Rel positively regulates *frzb* expression. **a** qPCR analysis for expression of Wnt antagonists (*frzb*, *sfrp1a*, *dkk1b* and *notum1a*) in dome stage embryos injected with ctrl MO or *rel* MO. Normalized values are shown as means ± SEM. *n* = 3, biologically independent samples. *P*-values from unpaired two-tailed *t*-tests are indicated. **b** WISH for *frzb* in dome stage embryos injected with ctrl MO or *rel* MO, animal view. **c, d** Rel inhibits Wnt/β-catenin signaling through Frzb. WISH for (**c**) Wnt targets *axin2* and *sp5l*; (**d**) *dharma* and *chordin* in embryos injected with ctrl MO, *rel* MO or *rel* MO with *frzb* mRNA at indicated stage, animal view. Box plots of the angle of marker genes show first and third quartile, median is represented by a line, whiskers indicate the minimum and maximum. Each dot represents one embryo. *P*-values from two-tailed one-way ANOVAs with Sidak correction are indicated. **e, f** Rel activates promoter of *frzb*. **e** Top panel shows the schematic diagram of the upstream region of zebrafish *frzb* gene. The NFκB-binding element possessing the strongest potential to bind to Rel homodimer is marked with red "pin" while others are marked by black "pins". Gray and blue boxes indicate Exons and UTRs, respectively. Bottom panel shows the schematic diagrams of the reporter constructs, frzb:luc and frzb:luc MT. **f** FISH for *luciferase* (magenta) and *gfp* (green) in dome-stage embryos injected with pDha-GFP and frzb:luc or frzb:luc MT, with MOs as indicated, animal view. Scale bar = 200 μm. Source data are provided as a Source Data file.

the hidden roles of Rel during DV axis formation by the combinatorial analyses. *rel* knockout mutants generated using CRISPR/Cas9 have no significant embryonic defects, whereas MO-mediated knockdown and in vivo imaging analyses revealed the involvement of *rel* in dorsal organizer formation. The upregulation of another *rel* homolog, *rela*, compensates for the loss of *rel* in the mutant. Thus, by taking advantage of knockdown, knockout, and imaging, we succeeded in discovering the previously unknown function and regulation of NFκB in vertebrate axis formation. Our finding is consistent with the previous *Xenopus* studies showing that overexpression of Xrel3 or XrelA (homolog of zebrafish Rel or Rela, respectively) inhibited the dorsal formation and the dorsal marker gene expression[20,21], which indicates that endogenous Xrel3 and XrelA may play roles similar to zebrafish Rel and Rela in *Xenopus* organizer formation.

The present study shows that Tlr/NFκB activates the transcription of Frzb to inhibit canonical Wnt ligands. Notably, the *frzb* promoter region of other vertebrates contains several potential NFκB-binding elements, including in *Xenopus*, chicken, and mouse (Supplementary Fig. 3e), which also generate organizer structures through Wnt/β-catenin signaling[1,69–72]. This suggests the potentially conserved roles of the Tlr-NFκB-Frzb-Wnt axis in organizer formation across widely different taxa.

Interestingly, whereas dorsally activated Wnt/β-catenin signaling in zebrafish initiates DV axis formation and negatively regulates itself through Tlr/NFκB-mediated indirect induction of the Wnt antagonist Frzb, ventrally activated Tlr/NFκB primes DV patterning and negatively control its activity through direct induction of WntD expression in *Drosophila* (Fig. 7). This ventral Tlr/NFκB-driven DV patterning is observed not only in *Drosophila* but also in other insects, such as *Tribolium* and *Gryllus*[73–75]. Thus, the roles of Tlr/NFκB have changed in the course of evolution. This change seems to be associated with the appearance of Wnt/β-catenin signaling-mediated organizer formation and NFκB-mediated Wnt antagonist (Frzb) induction. In fact, Wnt/β-catenin signaling is required for the formation of the Chordin-secreting organizer in vertebrates, amphioxus (a cephalochordate)[76], and the region functionally equivalent to the organizer in sea urchins (echinoderms)[77,78], whereas Wnt/β-catenin activity is not involved in the initial induction of the signaling centre organizing DV patterning in insects including *Drosophila*. Furthermore, the Sfrp3/4 subfamily of Wnt antagonist genes, to which *frzb* (also called Sfrp3) belongs[79], is found in the genome of vertebrates, cephalochordates, and echinoderms, which possess the organizer or related structures[76,78], but not in that of insects and nematodes, which do not[80]. It may be concluded that the evolutionary acquisition of organizer formation mediated by Wnt and its antagonist (Frzb) enabled vertebrates, cephalochordates, and echinoderms to change the roles of Tlr/NFκB from 'initiator' to 'mediator between Wnt and Frzb' in DV patterning.

We also demonstrate that the Toll-like receptor Tlr4 mediates Rel activation downstream of Wnt/β-catenin signaling during organizer formation using a specific inhibitor and a dominant-negative mutant. Although Tlr4 was first identified as a receptor of LPS derived from gram-negative bacteria[50,67], many endogenous Tlr4 ligands, including hyaluronan and fibronectin, have also been identified[81,82]. A recent study using primary cell cultures and in vitro binding experiments also showed that Wnt5 can directly bind to Tlr4 to stimulate NFκB activation[53]. Previous zebrafish studies have shown that zygotic expression of *wnt5b* is induced during the organizer formation stage[6] and that maternal-zygotic *wnt5b* mutants exhibit organizer size expansion[57] like Tlr4-inhibited embryos and *rel* morphants. The Wnt5 subclass is called a non-canonical Wnt ligand and negatively regulates β-catenin signaling-mediated dorsal formation by activating the $Ca^{2+}$ pathway[54–57]. However, the mechanisms controlling *wnt5b* expression in early zebrafish embryos and the role of Wnt5 as a Tlr4 ligand in vivo remain unclear. Here, we demonstrate that Wnt/β-catenin signaling stimulates *wnt5b* expression in developing organizer and that Wnt5b acts as a potential Tlr4 ligand to block Wnt/β-catenin signaling. Thus, our study revealed previously unidentified interactions between the canonical and non-canonical Wnt signaling pathways.

Although several transgenic NFκB reporter lines have been generated in zebrafish and mice, the activation of NFκB was not detected in early embryos around the initial stage of DV axis formation[83–85]. In this study, we successfully detected spatiotemporal activation of NFκB in early embryos using a transgenic zebrafish reporter. Interestingly, NFκB activation was detected as a "salt and pepper pattern" from 3.7 hpf and then restricted to the dorsal region from 4.3 hpf (Fig. 1b). The significance of the earliest salt-and-pepper pattern is still unclear. Although germ cells also showed a dotty pattern in early zebrafish embryos[86,87], they did not express NFκB reporter (Supplementary Fig. 1d), indicating that NFκB is activated in other cells. It is important to investigate this NFκB activation to fully understand the roles of endogenous NFκB during vertebrate embryogenesis.

Bacterial infections during pregnancy have been linked to various negative pregnancy outcomes, including spontaneous abortion, premature birth, stillbirth, intrauterine growth restriction, and fetal neurological defects in humans[88–90]. Tlr4 is a well-known sensor that recognizes the presence of bacterial infection[91,92]. A previous study reported that early mouse embryos express Tlr4[93,94], and maternal LPS elevated by bacterial infection negatively affects mouse fetal development and induces intra-uterine fetal death[95]. However, the mechanisms by which bacterial infection affects embryonic development are not fully understood. The current consensus is that inflammatory cytokines and chemokines, which are small immunological proteins, likely play a central role in infection-associated preterm birth and fetal injury[90,95]. In our study, we found that LPS severely interrupts zebrafish embryonic axis formation through Tlr4/NFκB signaling, which provides a previously undescribed mechanism of bacteria-induced defects in embryogenesis. We expect that our study using zebrafish as a model can deepen insights into the mechanisms by which infection influences early embryonic development.

## Methods
### Ethical approval
All experimental animal care was performed in accordance with the institutional and national guidelines and regulations. The study

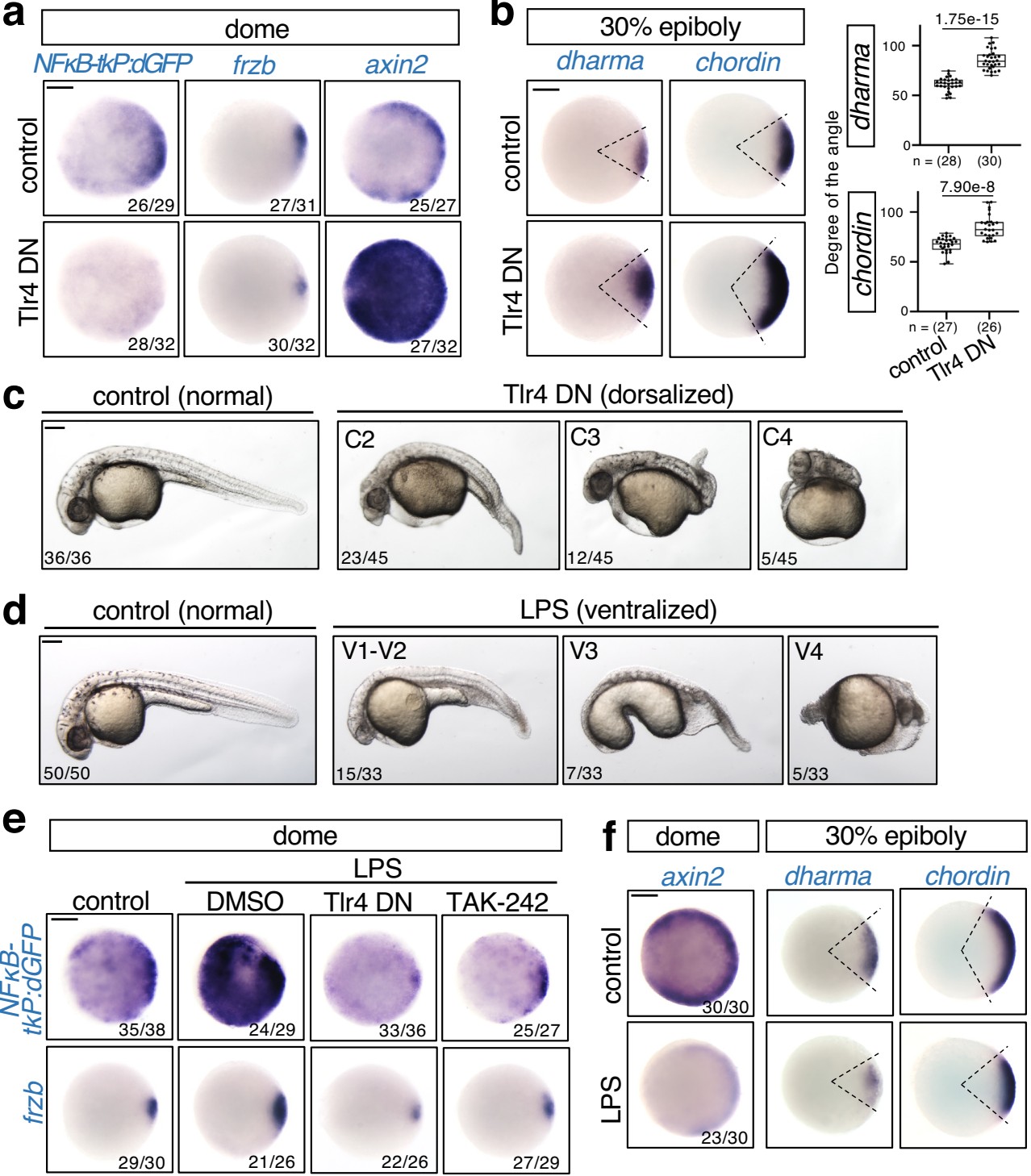

**Fig. 5 | Tlr4 activates NFκB to stimulate *frzb*-mediated restriction of dorsal organizer formation. a**–**c** Inhibition of Tlr4 reduces *frzb* expression and enhances Wnt/β-catenin signaling and dorsal organizer formation. Embryos were injected with mKO2 (control) or Tlr4 DN mRNA. WISH for (**a**) *dGFP* in NFκB-tkP:dGFP-transgenic; *frzb* and *axin2* in WT, (**b**) *dharma* and *chordin* in WT embryos at the indicated stage. Animal views. Box plots of the angle of marker genes show first and third quartile, median is represented by a line, whiskers indicate the minimum and maximum. Each dot represents one embryo. *P*-values from unpaired two-tailed *t*-tests are indicated. **c** Representative pictures of 27 hpf larvae, lateral views with anterior to the left. The strength of dorsalization was scored. **d**–**f** Forced activation of Tlr4 by injection of lipopolysaccharide (LPS) activates NFκB signaling and inhibits dorsal organizer formation. **d** Phenotypes of 27 hpf larvae injected with LPS, uninjected as control. The strength of ventralization was scored. Lateral views with anterior to the left. **e** Embryos were injected with LPS and treated with DMSO or TAK-242 or co-injected with Tlr4 DN, uninjected as control. DMSO and TAK-242 were treated from 3 hpf to dome stage. WISH for *dGFP* in NFκB-tkP:dGFP-transgenic; *frzb* in WT embryos at dome stage. **f** WISH for *axin2*, *dharma* and *chordin* in embryos injected with LPS, uninjected as control. Animal views. Scale bar = 200 μm. Source data are provided as a Source Data file.

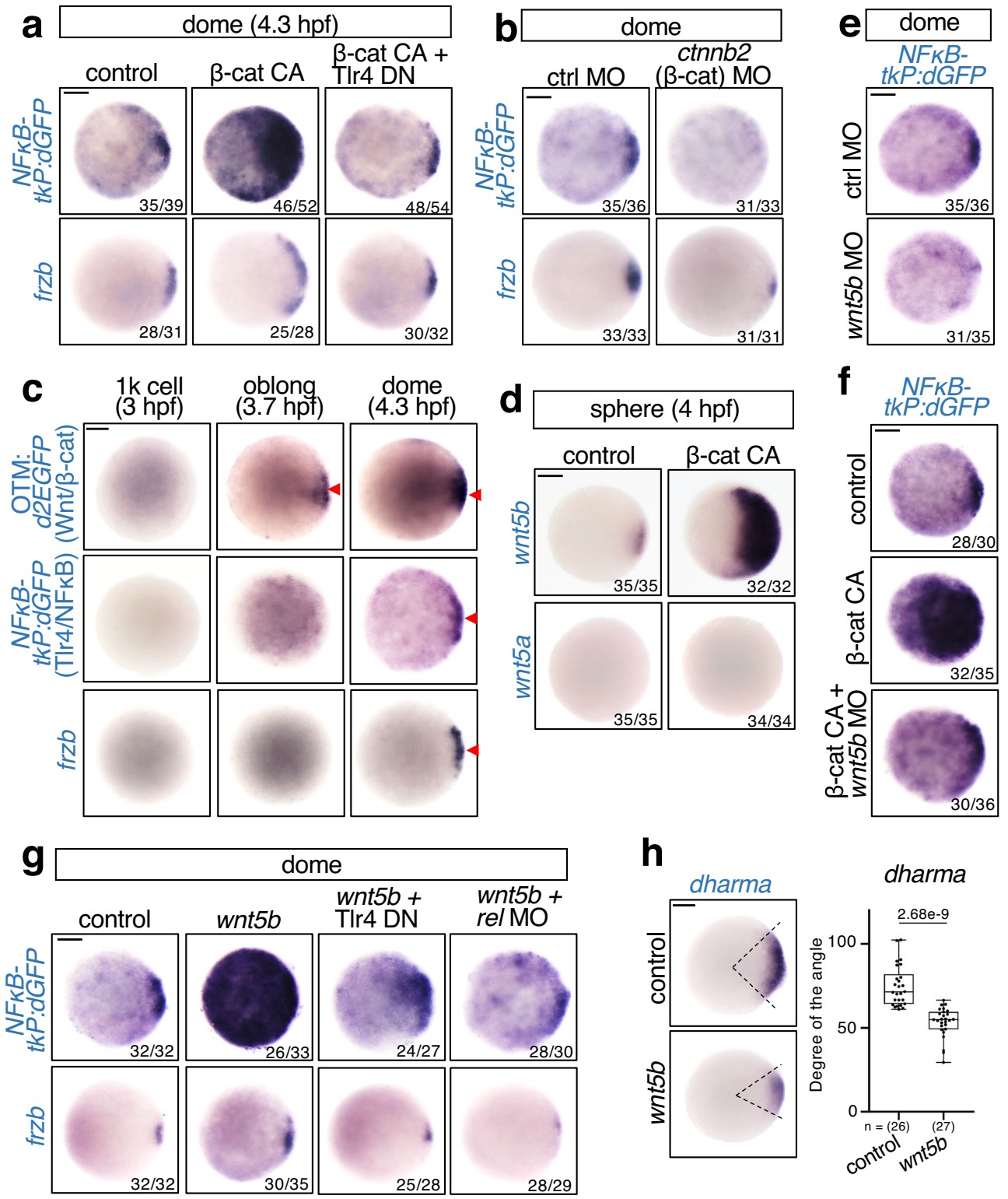

protocol was approved by the Institutional Animal Care and Use Committee of Osaka University (RIMD Permit# R02-04). The study was conducted according to the ARRIVE guidelines.

**Zebrafish maintenance**

Zebrafish Wild-type strain (AB), Tg(OTM:d2EGFP)[52], Tg(NFκB-tkP:dGFP), and *rel* mutant were raised and maintained under standard conditions. Tg(NFκB-tkP:dGFP) and *rel* mutant were generated in this study. Assays were conducted in zebrafish embryos and

larvae at 3–27 hpf. At these developmental stages, sex is not yet determined. All experimental animal care was performed in accordance with the institutional and national guidelines and regulations.

**Reporter plasmid construction**

To produce the reporter plasmid NFκB-tkP:dGFP, the Tcf/Lef-binding sequence and the minimal promoter of the OTM:d2EGFP (Tcf/Lef-miniP:dGFP) plasmid[52] was replaced with six copies of consensus NFκB

**Fig. 6 | β-catenin stimulates Wnt5b-mediated Tlr4/NFκB activation. a, b** β-catenin activates NFκB via Tlr4. **a** Embryos were injected with control (mKO2) or constitutively active β-catenin mutant (β-cat CA) mRNA, with or without Tlr4 DN mRNA. **b** Embryos were injected with control or *ctnnb2* (zebrafish β-catenin) MO. WISH for *dGFP* in NFκB-tkP:dGFP-transgenic; *frzb* in WT embryos at the dome stage, animal view. Scale bar = 200 μm. **c** NFκB is activated following Wnt/β-catenin activation in the developing dorsal organizer. WISH for *dGFP* in OTM:d2EGFP and NFκB-tkP:dGFP-transgenic embryos and *frzb* in WT embryos at the indicated stage. The *dGFP*-expressing and *frzb*-expressing dorsal regions are indicated by red arrowheads. Scale bar = 200 μm. **d** β-catenin activates *wnt5b* expression in early embryos. Embryos were injected with control (mKO2) or constitutively active β-catenin mutant (β-catCA) mRNA. WISH for *wnt5a* and *wnt5b* at the sphere stage, animal view. Scale bar = 200 μm. **e, f** Wnt/β-catenin signaling activates NFκB via

Wnt5b. **e** Embryos were injected with ctrl or *wnt5b* MO. **f** Embryos were injected with control (mKO2) or constitutively active β-catenin mutant (β-cat CA) mRNA, with or without *wnt5b* MO. WISH for *dGFP* in NFκB-tkP:dGFP-transgenic embryos at the dome stage, animal view. Scale bar = 200 μm. **g, h** Wnt5b activates *frzb* expression through Tlr4/Rel. **g** Embryos were injected with control (mKO2) or *wnt5b* mRNA, and co-injected with Tlr4 DN mRNA or *rel* MO. WISH for *dGFP* in NFκB-tkP:dGFP-transgenic, *frzb* in WT embryos at the dome stage, animal view. Scale bar = 200 μm. **h** WISH for *dharma* in WT embryos injected with control (mKO2) or *wnt5b* mRNA at 30% epiboly stage. Scale bar = 200 μm. Box plots of the angle of marker gene show first and third quartile, median is represented by a line, whiskers indicate the minimum and maximum. Each dot represents one embryo. *P*-values for unpaired two-tailed t-tests are indicated. Source data are provided as a Source Data file.

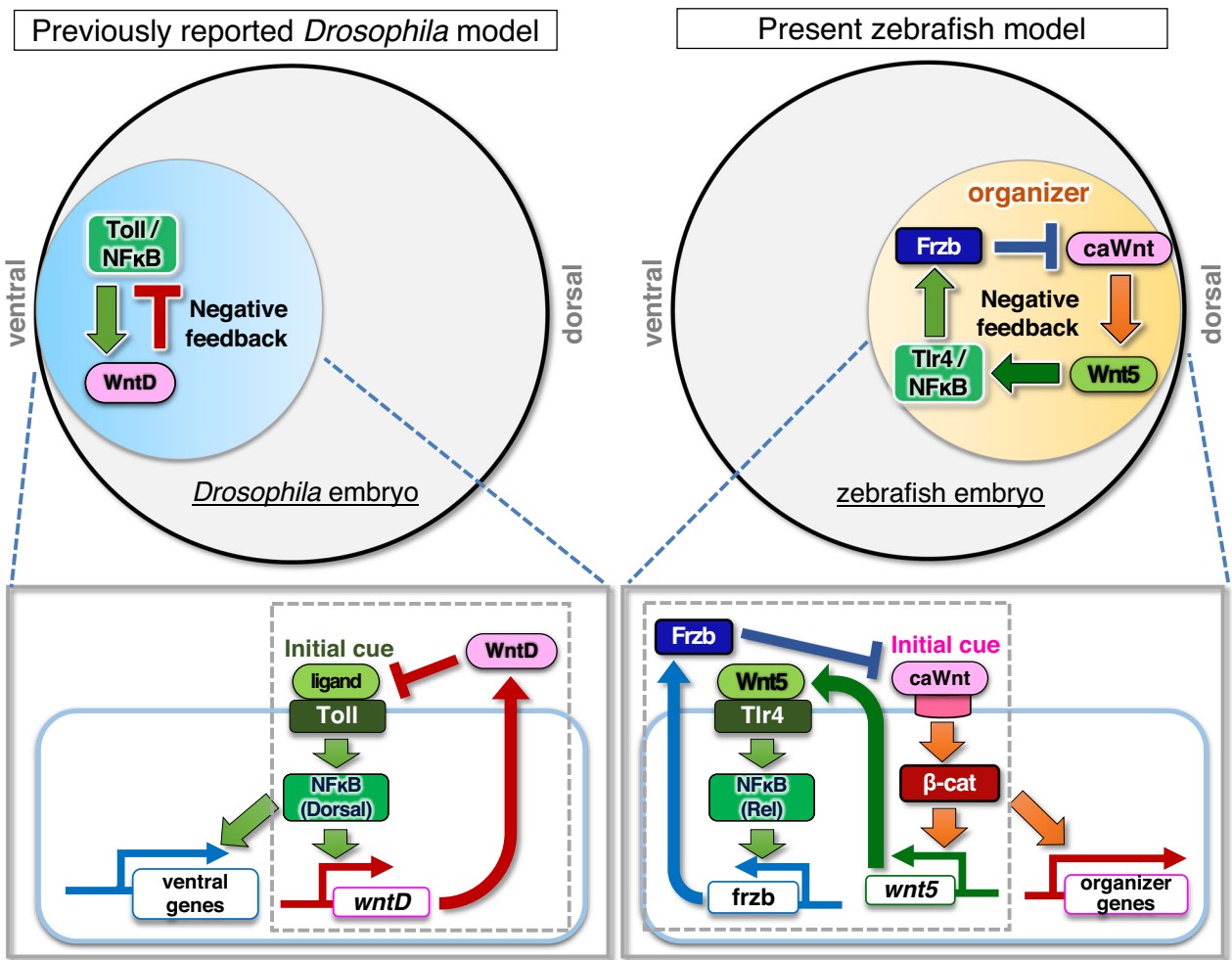

**Fig. 7 | Negative feedback loop between canonical/non-canonical Wnts and Tlr4/NFκB determines the precise size of zebrafish dorsal organizer.** Model of the role of Tlr/NFκB signaling in the initiation of embryonic DV axis formation. In *Drosophila*, Wnt-mediated negative feedback regulation of Toll/NFκB signaling

determines the embryonic DV pattern. In zebrafish, negative feedback regulation between canonical Wnt (caWnt), non-canonical Wnt (Wnt5b), and Tlr4/NFκB generates the precise size of the dorsal organizer and the consequent DV pattern.

binding sites (CTCGAGCGGAAAGTCCCACGGAAAGTCCCACGGAAAG TCCCACGGAAAGTCCCACGGAAAGTCCCACGGAAAGTCCCA)[96] and a thymidine kinase (tk) minimal promoter (tkP: CCCGCCCAGCGTCTT GTCATTGGCGAATTCGAACACGCAGATGCAGTCGGGGCGGCGCGGTC CCAGGTCCACTTCGCATATTAAGGTGACGCGTGTGGCCTCGAACACC GAGCGACCCTGCAG). The NFκB-binding element search tool (http://thebrain.bwh.harvard.edu/nfkb/) predicted that all NFκB family members could bind to these consensus NFκB binding sites on the reporter gene.

## Generation of transgenic and mutant zebrafish

For NFκB-tkP:dGFP transgenic zebrafish, 50 pg reporter plasmid DNA with 25 pg Tol2 transposase mRNA was injected into one-cell-stage wild-type zebrafish (AB) embryos. Transgenic fish that strongly expressed dGFP were outcrossed with wild-type fish to produce a transgenic line carrying a single transgene, Tg (NFκB-tkP:dGFP). This line was maintained as homozygous transgenic fish.

For CRISPR/Cas9-mediated generation of *rel* knockout mutant zebrafish, gRNA design and in vitro transcription of Cas9 RNA were

performed according to previously reported protocols[97]. Cas9 mRNA and gRNA were synthesized in vitro and co-injected into one-cell-stage wild-type zebrafish (AB) embryos. The sequence of the selected CRISPR target site was 5′-CGTTCTGCGGGCAGCATACC**AGG**−3′ (bold lettering indicates the PAM motif). Adult F0 fish were outcrossed with AB WT fish, and DNA was extracted from F1 progeny. Mutations were identified through direct sequencing of the PCR amplicon comprising the CRISPR target region.

## Plasmids

The zebrafish *iκbαb, rel, frzb, sfrp1a*, and *wnt5b* coding sequences were amplified using a zebrafish cDNA library. The mKO2 cDNA was purchased from MBL (Tokyo, Japan). *rel*5′ untranslated regions were annealed by DNA oligos, and cDNA for mKO2 was PCR-amplified. These two DNA fragments were cloned into the multi-cloning site of the pCS2p+ vector using the In-Fusion® HD Cloning Kit (Takara, Kusatsu, Japan). The dominant negative form of Tlr4 was PCR-amplified from mouse Tlr4 (Addgene #13085) with deletion of the TIR domain[47]. The N-terminus truncated mouse β-catenin (β-cat CA) has been described previously[98].

To prepare *frzb*-luciferase plasmid, the *frzb* gene upstream region (−500-1), including a putative NFκB binding site (TGTAGTTTCC), was amplified from a zebrafish genomic library and inserted upstream of the firefly luciferase gene in the pGL4 vector (Promega, Madison, WI, USA). A mutant reporter, in which a putative NFκB-binding site was deleted, was generated using the QuikChange Site-Directed Mutagenesis Kit (#210518; Agilent, Santa Clara, CA, USA). The pDha-1420GFP plasmid has been previously described[45].

## Injection of mRNA, plasmid, morpholino, and LPS

Capped mRNA was synthesized using the SP6 mMessage mMachine Kit (Ambion, Austin, TX, USA) and purified using Micro Bio-Spin columns (Bio-Rad, Hercules, CA, USA).

We injected synthesized mRNA (50 pg of *iκbαb*, 10 pg of *rel*, 500 pg of *frzb*, 500 pg of *sfrp1a*, 2 ng of Tlr4 DN, 20 pg of β-cat CA, and 120 pg of *wnt5b*) at the one-cell stage of zebrafish embryos.

Antisense oligo MOs (Gene Tools, Philomath, OR, USA) were injected into one-cell-stage embryos. Translation-blocking morpholinos used were *rel* MO (5 ng), *rela* MO (3 ng)[34], *frzb* MO (10 ng)[6], *ctnnb2* MO (10 ng)[51], *wnt5b* MO (8 ng)[58] and standard control (3–10 ng). MO sequences are shown in Supplementary Table 1.

LPS (12 ng) from *Escherichia coli* O111:B4 (L2630; Sigma-Aldrich, St. Louis, MO, USA) was injected at the single-cell stage of the embryos.

## RNA probe synthesis and whole-mount in situ hybridization

RNA probes were generated using gene-specific sequences cloned into multi-cloning sites of pBluescript SK+, pCS2p+, or pCRII-TOPO vectors (Thermo Fisher Scientific, Waltham, MA, USA). Linearized templates were subjected to in vitro transcription with DIG- or FITC-conjugated NTP (Sigma-Aldrich) using T3 (Promega) T7 or SP6 RNA polymerase (Takara), and then purified with RNA Clean & Concentrator Kits (Zymo Research, Irvine, CA, USA). The probe for *chordin* was preciously described[28]. For *rela*, the cDNA cloned into the pCR4-TOPO vector was purchased from TransOMIC (Huntsville, AL, USA). The Xho1-digested plasmid was subjected to in vitro transcription using T7 polymerase. Primer sequences used for cloning of other probes are shown in Supplementary Table 2.

Whole-mount in situ hybridization was performed according to a standard protocol. Fluorescence in situ hybridization was performed according to a previously described protocol[99]. In brief, embryos were fixed by 4% paraformaldehyde at 4 °C overnight and dehydrated in methanol at −20 °C overnight. Then, embryos were hybridized with FITC- or DIG-labeled RNA probes in Pre-Hyb solution (50% formamide, 5X SSC, 100 μg/ml yeast RNA, 50 μg/ml Heparin, 0.25% tween-20, 0.01 M Citric acid, pH 6.0–6.5) overnight at 68 °C, followed by stringent washes. Embryos were incubated with anti-Fluor-POD or anti-DIG-POD (Roche, Basel, Switzerland) overnight at 4 °C and then incubated with FITC-, Cy3-, or Cy5-tyramide (Akoya Biosciences, Marlborough, MA, USA). Images were taken using an M205A stereomicroscope (Leica Microsystems, Wetzlar, Germany) and an FV3000 confocal laser scanning microscope (Evident, Tokyo, Japan).

## Luciferase assay in zebrafish embryos

Frzb-luc plasmid (30 pg) was injected into one-cell-stage zebrafish embryos to detect *frzb* promoter activity. Fluorescent in situ hybridization (FISH) for *luciferase* was performed according to a previously described protocol[99]. Images were taken using a M205A stereomicroscope.

## Genomic DNA isolation and Southern blot analysis

The tail fins of adult transgenic fish were amputated using a razor and transferred to a lysis buffer containing 0.1 μg/μl Proteinase K (ProK). The samples were incubated overnight at 55 °C, followed by standard ethanol precipitation. Purified genomic DNA samples were digested with EcoRI, which cuts the plasmid reporter. Southern blot hybridization was performed using a digoxigenin (Roche, Basel, Switzerland)-labeled probe and standard methods.

## Quantitative PCR

Zebrafish embryos were randomly collected. Total RNA content from 25 embryos was purified using TRIzol reagent (Invitrogen), and cDNA was synthesized using ReverTra Ace qPCR RT Master Mix (Toyobo, Osaka, Japan). Quantitative PCR (qPCR) was performed on an Mx3000P QPCR system (Agilent Technologies) with THUNDERBIRD SYBR qPCR Mix (Toyobo), and qPCR was performed in triplicate. The levels of *actb1, ef1α*, and *rpl13* were used as a loading control. qPCR cycling conditions were as follows: 95 °C for 1 min, [95 °C for 10 s, and 60 °C for 30 s] (45 cycles), followed by dissociation curve analysis. The primer list is shown in Supplementary Table 1.

## Cell culture and transfection

HEK293 cells (#CRL-1573™, ATCC) were grown in Dulbecco's modified Eagle's medium (DMEM) supplemented with 10% foetal bovine serum (FBS) and 100 U/ml penicillin-streptomycin (Nacalai Tesque, Kyoto, Japan). Cells were transfected with expression plasmids encoding zebrafish Rel, mouse c-Rel, and the NFκB signaling activators TAB1 and TAK1[100] and reporter plasmids using polyethyleneimine MW 25000 (Polysciences, Warrington, PA, USA).

## Chemical treatment

TAK-242 (243984-11-4; Cayman Chemical, Ann Arbor, MI, USA) was dissolved in dimethyl sulfoxide (DMSO) at 10 mg/ml and stored at −30 °C. TAK-242 (40 μM) was treated from 3 hpf. To observe the phenotype at 27 hpf, embryos were washed thrice with egg water at 4.7 hpf to remove TAK-242.

## Statistics and reproducibility

Statistical analyses were performed using GraphPad Prism software v8.0.1 (GraphPad Software, Boston, MA, USA). The statistical significance of differences between groups in all datasets was calculated using a two-tailed unpaired t-test or one-way analysis of variance. *P*-values ≤ 0.05 were considered statistically significant. Figures of representative images or plots were reproduced in at least two (Figs. 4b–d, f, 5a–f, and 6a–h and Supplementary Figs. 1a–d, 2d, 2f, 3c, d, 4a–e, 5a, b, and6a), or three or more (Figs. 1a–i, 2a–e, 3a–d and 4a and Supplementary Figs. 2b, c, e, 2g, i, 3a, b, 5c–f, and 6b) independent experiments.

No statistical method was used to predetermine the sample size. No data were excluded from the analyses. Animals were randomly assigned to each experimental group. The Investigators were not blinded to allocation during experiments and outcome assessment.

**Reporting summary**

Further information on research design is available in the Nature Portfolio Reporting Summary linked to this article.

## Data availability

All the data supporting this study are available within the article, supplementary information, and source data. Source data are provided with this paper.

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

## Acknowledgements

We thank M. Hibi for providing plasmids and helpful discussions and A. Kawahara, T Masuda, and Ishitani lab members for their helpful discussions, technical support, and fish maintenance. This research was supported by the Takeda Science Foundation (T.I.), Mitsubishi Foundation (T.I.), Daiichi Sankyo Foundation (T.I.), Uehara Memorial Foundation (T.I.), Mochida Memorial Foundation (T.I.), Ono Medical Foundation (T.I.), SECOM Science and Technology Foundation (T.I.), KOSE Cosmetology Foundation (T.I.), Naito Foundation (T.I.), JST FOREST (M.O.), Grant-in-Aid for Transformative Research Areas(A) (21H05287) (T.I.), Scientific Research (B) (22H02820) (T.I.), Challenging Exploratory Research (23K18242) (T.I.)., Transformative Research Areas(B) (20H05791) (M.O.), and JSPS Fellows (21J14254) (J.Z.).

## Author contributions

Conception and design: J.Z. and T.I.; Investigation: J.Z., S.A., S.O., S.I., and M.O.; Writing and review: J.Z. and T.I.; Writing contribution and review: S.A., S.O., S.I., and M.O.

## Competing interests

The authors declare no competing interests.
