## [Peer Review File · Nature Communications]

Determining zebrafish dorsal organizer size by a negative feedback loop between canonical/non-canonical Wnts and Tlr4/NF κ BREVIEWER COMMENTS

Reviewer #1 (Remarks to the Author):

This is a very interesting study that provides significant novel mechanistic insight into a fundamental biological question of broad interest, namely the specification of the dorsoventral body axis and the organizer in vertebrate embryos. Excitingly, it shows that NfκB signaling, which is important for very early symmetry breaking and specification of the ventral side in *Drosophila*, also plays a role in dorsoventral specification in early zebrafish embryos. This adds a highly intriguing layer to the previous stunning findings that the BMP-BMP inhibitor molecular system regulating dorsoventral axis specification is conserved between flies and vertebrates despite the orientation of the body axes have reverted during evolution. The fact that this principle seems to extend also to NfκB, which in flies specifies the ventral side, but in vertebrates is active on the dorsal side, is very interesting news.

The authors also present an intriguing mechanism by which NfκB acts: via activation of a noncanonical Wnt ligand which then curbs Wnt/beta-catenin signaling; thus NfκB signaling restricts organizer size.

The study is very thoroughly conducted, the data are generally of very high quality and very well presented. Nevertheless, I do have a few criticism that the authors should address to strengthen their conclusions. If they can clarify these issue, I strongly recommend publication in *Nat. Comm.*

Major issues:

1. Fig 1D, 1F, 2B, 2D, 4D, 5A, S5B, 5E, 6E: data should be quantified by measuring the arc (angle) of dharma, chordin and vent marker gene expression, data should be plotted with variances, and statistical tests should be used to determine significance.
2. Fig. 1F: these data should be quantified, e.g. by measuring the fluorescent intensities of the in situ signals.
3. Whenever authors describe dorsalized or ventralized phenotypes at later embryonic stages, e.g. in Fig. 1E, H, 2E and so forth, they should make use of the well-established classification scheme (V1-V5, C1-C5) introduced by Kishimoto et al., *Development* 124, 4457-4466 (1997).
4. The data supporting a model whereby rel functions through induction of frzb expression are pretty convincing but could be strengthened as follows:
 - a) Can rel overexpression induce (ectopic) frzb expression?
 - b) frzb overexpression can rescue relMO-induced phenotypes. It would be interesting to test whether the other Wnt antagonists (in particular sfrp1a, which I guess is the one considered most important by the Thisses in their PNAS paper, doi: 10.1073/pnas.1106801108.) cannot. Certainly, in gain-of-function experiments it might not matter by which means the Wnt pathway is inhibited, and thus finding that sfrp1a also rescues rel MO would not falsify the model. Yet, finding that frzb can rescue but sfrp1a cannot would strengthen the model.
 - c) can rel overexpression induce (ectopic) expression of the frzb:Luciferase reporter?
5. Fig. S5B: data showing that the Tlr4 inhibitor reduces Nf-κB reporter and frzb expression is not convincing. It should be supported by quantification, e.g. by qRT-PCR.

6. Fig.S5B: Drug treatment regime/duration should be indicated in figure or described in legend.

7. Fig. 5D & 6E: the frzb expression changes in these panels are too subtle to be fully conclusive. Can they be supported by quantification, e.g. by qRT-PCR?

8. The data indicating that Wnt5b activates Nf-kappaB signaling, and that it itself is activated by beta-catenin are pretty convincing, but could be strengthened as follows:

a) can Wnt5b loss-of-function(e.g by morpholino) be shown to reduce activation of Nf-kappaB signaling (reporter expression)?

b) can Wnt5b be shown to be required for the activation of NF-kappaB signaling by gain-of-canonical Wnt signaling (e.g. by overexpression of ca-beta-catenin?)

9. The new reporter transgene should be submitted to ZFIN to receive a name backed by official nomenclature and an allele number.

10. qPCRs were normalized to only 1 reference gene, which is not according to current standards. Rather, normalization should be done relative to the (geometric) mean of at least two standards. I realize that it's asking a lot of the authors to repeat all of their qRT-PCRs. Maybe they can at least repeat one crucial experiment using 2 references and show that they also get significant results?

Minor:

1. Introduction line 61. Authors state that Wnt8 (amongst others) would activate beta-catenin signaling in vertebrates to mediate dorsal axis specification. The papers they cite do not support this (Tao et al 2005 showed that it's Wnt11 in frogs). They should rephrase to make it clear that it's likely different Wnts in different species and cite Bernard and Christine Thisse who showed that it's Wnt8a in zebrafish. doi: 10.1073/pnas.1106801108

2. Line 65: "Dorsal" should not be in capital here, since capital "Dorsal" is the Drosophila gene.

3. Fig. S1 legends: authors should describe in the legend whether the reporter was also transiently transfected or whether stable cells were used.

Reviewer #2 (Remarks to the Author):

AP and DV body axis formation is one of the most fundamental events in embryonic development of all bilaterally symmetrically organized animals. It has been shown for AP formation (Wnt/Dkk) and DV axis formation (BMP/Chordin in vertebrates and dpp/sog in insects) that negative feedback loops play an important role. A new negative feedback loop is described in the present work on pattern formation in fish, which brings together both signaling pathways of AP and DV axis formation. This negative feedback loop between canonical and noncanonical Wnts determines zebrafish organizer size, and it also involves Toll-like receptor (Tlr) /NFkB-mediated activation of the Wnt antagonist frzb.

In insects (*Drosophila*), DV axis formation is initiated on the ventral side of the embryo by an extracellularly acting serine protease cascade. At its end, the Toll receptor (Tlr) is activated, allowing nuclear translocation of the NF κ B homolog Dorsal. Dorsal (NF κ B) activates the ventral specification genes. These include the Chordin homolog *sog*, an antagonist of the BMP homolog *dpp*, and the Wnt8 homolog WntD. WntD in *Drosophila* mediates negative inhibition of Dorsal (NF κ B) signaling and is thus required for proportion control on the ventral side of the *Drosophila* embryo. – In vertebrates (amphibians, fish), DV axis formation is initiated by canonical Wnt signaling (Wnt8) in the organizer. Here, expression of Chordin (*sog*) is induced, which acts as a BMP (*dpp*) antagonist. It has been shown that NF κ B (Dorsal) family genes are also involved in DV axis formation in *Xenopus*. In zebrafish, however, mutant screens failed to isolate Tlr/NF κ B-related inhibitors in the context of organizer formation.

The authors have now shown that the organizer acts as a signaling center whose size is regulated by canonical Wnt/ β -catenin signaling, thus indirectly controlling DV axis formation. Size regulation occurs through Tlr/NF κ B-mediated activation of the Wnt antagonist *frzb*. The authors show that Wnt/ β -catenin signaling during organizer formation stimulates Tlr4/NF κ B (Rel) activation specifically in future dorsal embryonic tissue. Surprisingly, this activation is mediated via the noncanonical Wnt5 ligand, which itself, however, is dependent on canonical Wnt8/ β -catenin signaling. These are interesting new data revealing a Wnt-mediated role of Tlr/NF κ B in DV patterning in vertebrates.

There are several suggestions / critical comments on the ms

(i) The abstract, introduction, and discussion are very much focused on the findings obtained in *Drosophila* on the role of Tlr/NF κ B activation in the formation of the DV axis. I propose to focus these three sections on vertebrates first and compare the main findings with those in *Drosophila*/insects later. It would also be helpful to expand the discussion to provide a more complete picture of the role of Tlr/NF κ B activation in DV axis formation in mammals and other insects, such as *Tribolium*. I also missed a discussion of the insightful work of S Roth (Sachs et al 2015, *Elife*; Stappert et al 2016, *Development*).

(ii) The naming of genes can be notoriously confusing. My suggestion is, wherever possible, use only one terminology, and only in comparison e.g., with *Drosophila* or *Danio* consult the names used there in comparison. When comparing the insects vertebrates, it is made even more difficult by the fact that the DV is inverted in both groups with the same ligands and antagonists (*dpp*/BMP, *sog*/Chordin etc).

(iii) The schemes in Figure 7 could be simpler and optimized. The two upper schemes should be devoid of cellular details and much simpler, and the two lower interaction schemes should better summarize common aspects (Tlr/NF κ B activation loop) and negative feedback loops (Wnts). This would also clarify open questions, e.g., whether Wnt8/WntD is secreted outward, e.g., into the extracellular space, in *Drosophila*.

(iv) The cited work on mammalian tlr4/Wnt5 interaction has been done in cell cultures and may have only limited relevance for the organismic context

(v) Typo in Fig. 4F.

(vi) Imaging of luc-reporter constructs should be described in the M&M.

Reviewer #3 (Remarks to the Author):

It is well established that canonical Wnt/ β -catenin signaling is required for organizer formation in the zebrafish embryo and that the organizer plays an essential role in dorsal ventral patterning. Here the authors describe a novel function for the NF κ B homolog, Rel, in establishing the size of the organizer. They show that Wnt/ β -catenin signaling induces expression of the non-canonical Wnt, Wnt5b, which in turn promotes Rel activation via Toll-like receptor 4. Subsequently, Rel initiates negative feedback regulation of Wnt/ β -catenin by activating transcription of *frzb*, a Wnt antagonist thereby helping to define the size of the organizer. Maternal-zygotic CRISPR/Cas9 Rel mutants are phenotypically normal due to transcriptional compensation by *Rela*. The findings are interesting, and the data support the conclusions. Some additional experiments and controls would strengthen the findings.

-In situ hybridization expression of both the reporter (Figure 1B) and *rel* (Figure S2B) shows punctate staining, which is particularly striking for the reporter. Could this expression be in germ cells? Expression of both the reporter and *rel* is not limited to the dorsal side—is *rel* ever restricted dorsally? What is the expression pattern of *rela*?

-RNA rescue experiments are lacking for the morpholino experiments. The control shown in Figure S2D—is no longer considered an acceptable control in the field, see: <https://doi.org/10.1371/journal.pgen.1007000>

-The data suggests that *rel* and *rela* function redundantly in the organizer. Co-injection of both morpholinos would add additional useful information that could further support this conclusion.

-in the Discussion some discussion about how the downstream targets changed over the course of evolution (*Drosophila* vs vertebrates) would be a nice addition. Also, more discussion of the results in *Xenopus* and how they relate to the findings in this work seems warranted. The *Xenopus* findings are dismissed as being overexpression experiments, though the authors also perform several over-expression experiments.

Minor Points

-1st sentence in introduction (line 43) change “variable” to “various”

-Last sentence on page 5, beginning: “Surprisingly, the non-canonical Wnt5 ligand mediated...” is not a complete sentence

-scale bars missing from images

-Figure C the oblong stage panel of NF κ B transgene expression appears to be the same embryo as in Figure 1B—if it is the case, then it should be indicated in the figure legend

-Methods do not include cloning primer sequences and details on preparation of the in situ probes

Manuscript ID: NCOMMS-23-03727-T

Authors: Zou et al.

Title: **Determining the precise size of the zebrafish dorsal organizer by a negative feedback loop between canonical/non-canonical Wnts and Tlr4/NFκB**

The manuscript has been revised in accordance with the comments raised by the three referees. Our responses to their comments follow below. Textual changes are indicated in red font in the revised manuscript.

Author responses to the comments of Reviewer #1

We thank Reviewer#1 for the careful and constructive review of our paper and for the positive comments on our findings. As indicated in the responses below, we have addressed all comments and suggestions in the revised version of the manuscript.

Major issues:

1. Fig 1D, 1F, 2B, 2D, 4D, 5A, S5B, 5E, 6E: data should be quantified by measuring the arc (angle) of *dharm*, *chordin* and *vent* marker gene expression, data should be plotted with variances, and statistical tests should be used to determine significance.

Response: Following Reviewer#1's suggestion, we measured the arc (angle) of *dharm*, *chordin* and *vent* marker gene expression, plotted these together with their variances, and performed statistical tests on the outcomes. These are shown in revised Fig. 1e, 1h, 2b, 2c, 4d, 5b, supplementary Fig. 5d, g and Fig. 6h (corresponding to original Fig. 1D, 1F, 2B, 2D, 4D, 5A, S5B, 5E and 6E, respectively).

2. Fig. 1F: these data should be quantified, e.g. by measuring the fluorescent intensities of the in situ signals.

Response: As Reviewer#1 suggested, the fluorescent intensities were measured, and we confirmed that *rel* overexpression significantly enhanced NFκB reporter activity (revised supplementary Fig. 1c [corresponding to original Fig. 1F]).

3. Whenever authors describe dorsalized or ventralized phenotypes at later embryonic stages, e.g. in Fig. 1E, H, 2E and so forth, they should make use of the well-established classification scheme (V1-V5, C1-C5) introduced by Kishimoto et al., *Development* 124, 4457-4466 (1997).

Response: Following Reviewer #1's suggestion, in revised Fig. 1f, 1i, 2d, 5c, 5d, and supplementary Fig. 4a, 5e (corresponding to original Fig. 1E, 1H, 2E, 5B, 5C, S4A and S5C respectively), we made use of the suggested classification scheme (V1-V5, C1-C5) (Mullin et al., *Development* 1996; Kishimoto et al., *Development* 1997) to describe the dorsalized or ventralized phenotypes at later embryonic stages.

4. The data supporting a model whereby *rel* functions through induction of *frzb* expression are pretty convincing but could be strengthened as follows:

a) Can *rel* overexpression induce (ectopic) *frzb* expression?

Response: We appreciate these helpful suggestions.

We carried out experiments to confirm that *rel* overexpression enhanced the expression of endogenous *frzb* gene and *frzb:luc* reporter in the dorsal area (revised supplementary Fig. 3c, d), indicating that *Rel* positively regulates *frzb* expression. Notably, *Rel* overexpression did not induce ectopic expression of *frzb* and *frzb:luc* in the ventral area, suggesting that there may be some factors inhibiting *Rel* activation or *frzb* expression in the ventral region or facilitating them in the dorsal region. Consistent with this idea, a *Rel* activator *Wnt5b* is specifically expressed in the dorsal margin of zebrafish embryos (revised Fig. 6d), and zebrafish *IκB* homologue *ikbab* is ubiquitously expressed in early embryos (<https://zfin.org/ZDB-FIG-050810-392>). These expressions would restrict *Rel*-mediated *frzb* induction to the dorsal area.

b) *frzb* overexpression can rescue *rel*MO-induced phenotypes. It would be interesting to test whether the other Wnt antagonists (in particular *sfrp1a*, which I guess is the one considered most important by the Thisses in their PNAS paper, doi: 10.1073/pnas.1106801108.) cannot. Certainly, in gain-of-function experiments it might not matter by which means the Wnt pathway is inhibited, and thus finding that *sfrp1a* also rescues *rel* MO would not falsify the model. Yet, finding that *frzb* can rescue but *sfrp1a* cannot would strengthen the model.

Response: Following the suggestion, we tested if *rel* MO-induced phenotypes can be

rescued by *sfrp1a* overexpression. As shown in revised supplementary Fig. 4d-e, overexpression of *sfrp1a* also rescued *rel* MO-induced activation of Wnt signalling and expansion of dorsal organizer, suggesting that negative regulation of Wnt is important for size control of the dorsal organizer.

On the other hand, our results reveal that Rel regulates the expression of *frzb*, but not other of Wnt antagonist (*sfrp1a*, *dkk1b*, and *notum1a*) during organizer formation (revised Fig. 4a, Supplementary Fig. 3b), suggesting that Frzb, but not other Wnt antagonists, plays an essential role in organizer formation downstream of Rel/NFκB. On the other hand, Sfrp1 may contribute to organizer formation in a Rel/NFκB-independent manner.

c) can rel overexpression induce (ectopic) expression of the *frzb*:Luciferase reporter?

Response: As we described above, we confirmed that *rel* overexpression enhances the expression of the *frzb*:luc reporter (revised supplementary Fig 4c). This result strengthens our model conception that Rel directly activates the transcription of *frzb*.

5. Fig. S5B: data showing that the Tlr4 inhibitor reduces Nf-kB reporter and *frzb* expression is not convincing. It should be supported by quantification, e.g. by qRT-PCR.

Response: As Reviewer#1 suggested, we performed qRT-PCR to validate this conclusion. As shown in revised supplementary Fig. 5c, treatment with Tlr4 inhibitor (TAK-242) significantly reduced the expression of NFκB reporter and *frzb*.

6. Fig.S5B: Drug treatment regime/duration should be indicated in figure or described in legend.

Response: As Reviewer#1 suggested, we have added the description of the drug treatment duration in the figure legend.

7. Fig. 5D & 6E: the *frzb* expression changes in these panels are too subtle to be fully conclusive. Can they be supported by quantification, e.g. by qRT-PCR?

Response: As Reviewer#1 suggested, we performed qRT-PCR related to the experiments shown in the original Fig. 5D and 6E (corresponding to revised Fig. 5e and 6g). LPS treatment significantly increased *frzb* expression, and this increase was blocked by expression of Tlr4 DN or treatment with Tlr4 inhibitor (revised supplementary Fig. 5f).

Wnt5b overexpression significantly increased *frzb* expression and Tlr4 DN and *rel* MO blocked this increase (revised supplementary Fig. 6b)

8. The data indicating that Wnt5b activates NfkappaB signaling, and that it itself is activated by beta-catenin are pretty convincing, but could be strengthened as follows:

a) can Wnt5b loss-of-function (e.g. by morpholino) be shown to reduce activation of NfkappaB signaling (reporter expression)?

Response: We appreciate Reviewer #1's helpful comments.

As shown in revised Fig. 6e, injection of *wnt5b* translation-blocking morpholino (Zhang et al., *Nat Commun* 2020) reduced the NFκB reporter activity, suggesting that Wnt5b promotes NFκB signalling in early zebrafish embryos.

b) can Wnt5b be shown to be required for the activation of NFkappaB signaling by gain-of- canonical Wnt signaling (e.g. by overexpression of ca-beta-catenin?)

Response: To answer this question, we tested whether *wnt5b* MO injection blocks β-cat CA (constitutively active β-catenin)-induced activation of NFκB reporter. As shown in the revised Fig. 6f, knockdown of *wnt5b* restored the β-cat CA-induced activation of NFκB signalling, suggesting that canonical Wnt signalling activates NFκB signalling through Wnt5b.

9. The new reporter transgene should be submitted to ZFIN to receive a name backed by official nomenclature and an allele number.

Response: As Reviewer#1 suggested, we have submitted the new reporter gene to ZFIN and received an official name and an allele number:

<https://zfin.org/ZDB-ALT-230614-2>

10. qPCRs were normalized to only 1 reference gene, which is not according to current standards. Rather, normalization should be done relative to the (geometric) mean of at least two standards. I realize that it's asking a lot of the authors to repeat all of their qRT-PCRs. Maybe they can at least repeat one crucial experiment using 2 references and show that they also get significant results?

Response: Following Reviewer#1's suggestion, we performed qPCR analysis using two

other standard genes: *efl α* and *rpl13*, which are widely used in zebrafish studies (Tang et al., *Acta Biochim Biophys Sin* 2007).

qPCR analyses using three standard genes show that *rel* MO significantly enhanced dorsal marker gene expression in WT but not MZ*rel* embryos (revised supplementary Fig. 2e [corresponding to original Fig. 2C]). In addition, as shown in the revised Fig. 3b, 4a, and supplementary Fig. 3a, *rel* MO significantly enhanced the expression of Wnt target genes but decreased that of *frzb*.

Minor:

1. Introduction line 61. Authors state that Wnt8 (amongst others) would activate beta-catenin signaling in vertebrates to mediate dorsal axis specification. The papers they cite do not support this (Tao et al 2005 showed that it's Wnt11 in frogs). They should rephrase to make it clear that it's likely different Wnts in different species and cite Bernard and Christine Thisse who showed that it's Wnt8a in zebrafish. doi: 10.1073/pnas.1106801108

Response: Following Reviewer#1's suggestion, we have rephrased this section of the Introduction and cited the paper (Lu et al., *PNAS* 2011) which showed that Wnt8a activates β -catenin signalling to mediate dorsal axis specification in zebrafish (page 3).

2. Line 65: "Dorsal" should not be in capital here, since capital "Dorsal" is the *Drosophila* gene.

Response: As suggested, we corrected "Dorsal" to "dorsal" in the main text.

3. Fig. S1 legends: authors should describe in the legend whether the reporter was also transiently transfected or whether stable cells were used.

Response: The reporter was indeed transiently transfected; we added the relevant description to the legend.

Reviewer #2 (Remarks to the Author):

We thank Reviewer#2 for the positive comments regarding the interesting finding of the negative feedback regulation between Wnt and Tlr/NFκB in DV patterning in vertebrates. We also appreciate the insightful comments and suggestions provided, particularly those on evolution of DV axis formation among species. As indicated in the responses below, we have considered all comments and suggestions in the revised version of the manuscript.

(i) The abstract, introduction, and discussion are very much focused on the findings obtained in *Drosophila* on the role of Tlr/NFκB activation in the formation of the DV axis. I propose to focus these three sections on vertebrates first and compare the main findings with those in *Drosophila*/insects later. It would also be helpful to expand the discussion to provide a more complete picture of the role of Tlr/NFκB activation in DV axis formation in mammals and other insects, such as *Tribolium*. I also missed a discussion of the insightful work of S Roth (Sachs et al 2015, *Elife*; Stappert et al 2016, *Development*).

Response: Thank you for these helpful suggestions. Following Reviewer#2's suggestions, we revised the Abstract, Introduction and Discussion sections (page 2, 3–4, 12–13). We focused on vertebrates first and developed this into a comparison with *Drosophila* later on. We also added some discussion about the evolution of DV axis formation (page 13).

(ii) The naming of genes can be notoriously confusing. My suggestion is, wherever possible, use only one terminology, and only in comparison e.g., with *Drosophila* or *Danio* consult the names used there in comparison. When comparing the insects vertebrates, it is made even more difficult by the fact that the DV is inverted in both groups with the same ligands and antagonists (dpp/BMP, sog/Chordin etc).

Response: Following Reviewer#2's suggestion, we revised the naming of genes, mainly using the gene names of vertebrates.

(iii) The schemes in Figure 7 could be simpler and optimized. The two upper schemes should be devoid of cellular details and much simpler, and the two lower interaction schemes should better summarize common aspects (Tlr/NFκB activation loop) and negative feedback loops (Wnts). This would also clarify open questions, e.g., whether

Wnt8/WntD is secreted outward, e.g., into the extracellular space, in *Drosophila*.

Response: Following Reviewer#2's suggestion, we modified Figure 7 accordingly.

(iv) The cited work on mammalian tlr4/Wnt5 interaction has been done in cell cultures and may have only limited relevance for the organismic context

Response: Reviewer#2 pointed out that a previous study demonstrating mammalian Tlr4/Wnt5 interaction may have only limited relevance in the organismic context (Mehmeti et al., *Commun Biol* 2019). While that study showed the Tlr4/Wnt5 interaction using cell culture and *in vitro* binding assays, our findings show that Wnt5 activates NFκB signalling through Tlr4 in zebrafish embryos, thus demonstrating that Tlr4/Wnt5 interaction play important roles *in vivo*. We thus suggest that this constitutes a useful discussion point, and have expanded on this aspect in the Discussion (page 14).

(v) Typo in Fig. 4F.

Response: This has been corrected accordingly (frzb:luc).

(vi) Imaging of luc-reporter constructs should be described in the M&M.

Response: Following this suggestion, we added a description of luc-reporter imaging in the Methods section (page 18–19).

Reviewer #3 (Remarks to the Author):

We thank Reviewer #3 for the positive comments on our findings and for providing important and concrete suggestions, which were helpful for improving our manuscript. As indicated in the responses below, we have carefully considered all comments and suggestions in the revised version of the manuscript.

-In situ hybridization expression of both the reporter (Figure 1B) and *rel* (Figure S2B) shows punctate staining, which is particularly striking for the reporter. Could this expression be in germ cells? Expression of both the reporter and *rel* is not limited to the dorsal side—is *rel* ever restricted dorsally? What is the expression pattern of *rela*?

Response: We appreciate Reviewer#3's helpful comments. To examine whether NFκB reporter is expressed in germ cells, we performed double fluorescent *in situ* hybridization (FISH) to examine the expression patterns of *dGFP* (reporter) and *nanos3* (germ cell marker) at the dome stage. As shown in Supplementary Fig. 1d, the *nanos3* expression did not merge with the *dGFP*, indicating that the punctate stained cells were not germ cells.

With respect to the punctate staining of *rel* (original Figure S2B), we must apologize for misleading image material; the observed features were not *rel*-expressing cells but accidental markings caused by the BM purple substrates. We performed *in situ* hybridization again and prepared new figures for the revised supplementary Fig. 2b. As shown in revised figure, both *rel* and *rela* are ubiquitously expressed at the oblong, dome and 50% epiboly stages, suggesting that the dorsally restricted NFκB reporter activation is due to dorsal specific activation, but not expression, of NFκB. In fact, we found that Wnt/β-catenin signalling activates Tlr4 through Wnt5b in the dorsal region, thus inducing dorsally restricted NFκB activation.

-RNA rescue experiments are lacking for the morpholino experiments. The control shown in Figure S2D—is no longer considered an acceptable control in the field, see: <https://doi.org/10.1371/journal.pgen.1007000>

Response: Following Reviewer#3's suggestion, we added RNA rescue experiments for morpholino in the revised supplementary Fig. 2f. As shown in the figure, overexpression

of morpholino-insensitive *rel* mRNA rescued *rel* MO-induced dorsal expansion, indicating that *rel* MO indeed specifically blocks the function of Rel.

-The data suggests that *rel* and *rela* function redundantly in the organizer. Co-injection of both morpholinos would add additional useful information that could further support this conclusion.

Response: As shown in revised Fig. 2b and 2d, *rela* morpholino (MO) did not affect dorsal gene expression or induce a significantly dorsalized phenotype in WT embryos, indicating that Rel does not restrict dorsal organizer formation in WT embryos. To further confirm this, we have added results of co-injection experiments of both MOs. As shown in the revised supplementary Fig. 2g, *rela* MO alone in WT embryos did not affect the expression of dorsal marker genes. Notably, injection of both *rel* and *rela* MOs did not significantly change of dorsal marker gene expression compared to *rel* MO alone, indicating that Rel but not Rel α is the main NF κ B that functions in dorsal organizer formation in WT embryos.

However, in the *MZrel* mutant, *rela* is upregulated to compensate for the loss of *rel*. Therefore, *rela* MO diminished genetic compensation in the *MZrel* mutant, thus inducing expansion of the dorsal organizer and consequent dorsalization (summarized in Fig. 2f). It can thus be concluded that Rel α plays essential roles in organizer formation in *MZrel* mutants but not in WT embryos.

-in the Discussion some discussion about how the downstream targets changed over the course of evolution (*Drosophila* vs vertebrates) would be a nice addition. Also, more discussion of the results in *Xenopus* and how they relate to the findings in this work seems warranted. The *Xenopus* findings are dismissed as being overexpression experiments, though the authors also perform several over-expression experiments.

Response: Thank you for this thoughtful comment. Following Reviewer#3's suggestion, we added some discussion about how the roles of Tlr/NF κ B have changed over the course of evolution (page 13) and address the previous *Xenopus* studies in detail (page 12–13).

Minor Points

-1st sentence in introduction (line 43) change “variable” to “various”

Response: This was revised accordingly.

-Last sentence on page 5, beginning: “Surprisingly, the non-canonical Wnt5 ligand mediated...” is not a complete sentence

Response: We accordingly revised “mediated” to “mediates” to correct this sentence.

-scale bars missing from images

-Figure 6C the oblong stage panel of NF κ B transgene expression appears to be the same embryo as in Figure 1B—if it is the case, then it should be indicated in the figure legend

Response: We apologize for the confusion. To avoid the use of misleading imagery, we created new photos of NF κ B reporter embryos for the revised Fig. 6c.

-Methods do not include cloning primer sequences and details on preparation of the *in situ* probes

Response: Following this suggestion, we added cloning primer sequence in Supplementary Table 2 and details on the preparation of the *in situ* probes in the Methods section (page 18).

REVIEWERS' COMMENTS

Reviewer #1 (Remarks to the Author):

The authors have done a great job in convincingly addressing all concerns I had raised. They added data of high quality that further support and expand their conclusions. I can now fully recommend publication in NatComm.

Reviewer #2 (Remarks to the Author):

The authors have carefully addressed all of the reviewers' concerns and suggestions. To this end, the authors have revised sections of the abstract, introduction and discussion. The paper now provides a clear picture of the role of Tlr/NFκB activation in DV axis formation in mammals and insects, based on new data on zebrafish. They have also revised the summary figure, which now convincingly shows the negative feedback loop between canonical/noncanonical Wnt and Toll/Tlr4-NFκB signalling in Drosophila and zebrafish. In summary, this is important work on the evolutionary and mechanistic aspects of axis formation and the feedback mechanisms of signalling pathways that operate in the formation of AP and DV axes.

Reviewer #3 (Remarks to the Author):

The revisions to the manuscript have strengthened the conclusions of this interesting work. All of my concerns have been addressed.